# Co-regulation and function of *FOXM1/ RHNO1* bidirectional genes in cancer

Carter J Barger[1], Linda Chee[1], Mustafa Albahrani[1], Catalina Munoz-Trujillo[1], Lidia Boghean[1], Connor Branick[1], Kunle Odunsi[2], Ronny Drapkin[3], Lee Zou[4], Adam R Karpf[1]*

[1]Eppley Institute for Cancer Research and Fred & Pamela Buffett Cancer Center, University of Nebraska Medical Center, Omaha, United States; [2]Departments of Gynecologic Oncology, Immunology, and Center for Immunotherapy, Roswell Park Comprehensive Cancer Center, Buffalo, United States; [3]Penn Ovarian Cancer Research Center, University of Pennsylvania Perelman School of Medicine, Philadelphia, United States; [4]Massachusetts General Hospital Cancer Center, Harvard Medical School, Charlestown, United States

**Abstract** The FOXM1 transcription factor is an oncoprotein and a top biomarker of poor prognosis in human cancer. Overexpression and activation of FOXM1 is frequent in high-grade serous carcinoma (HGSC), the most common and lethal form of human ovarian cancer, and is linked to copy number gains at chromosome 12p13.33. We show that *FOXM1* is co-amplified and co-expressed with *RHNO1*, a gene involved in the ATR-Chk1 signaling pathway that functions in the DNA replication stress response. We demonstrate that *FOXM1* and *RHNO1* are head-to-head (i.e., bidirectional) genes (BDG) regulated by a bidirectional promoter (BDP) (named F/R-BDP). FOXM1 and RHNO1 each promote oncogenic phenotypes in HGSC cells, including clonogenic growth, DNA homologous recombination repair, and poly-ADP ribosylase inhibitor resistance. FOXM1 and RHNO1 are one of the first examples of oncogenic BDG, and therapeutic targeting of FOXM1/ RHNO1 BDG is a potential therapeutic approach for ovarian and other cancers.

*For correspondence:
adam.karpf@unmc.edu

Competing interests: The authors declare that no competing interests exist.

## Introduction

The forkhead/winged helix domain transcription factor FOXM1 promotes cancer by transactivating genes with oncogenic potential (*Halasi and Gartel, 2013a*; *Kalathil et al., 2020*). Cell phenotypes promoted by FOXM1 include cell cycle transitions (*Costa, 2005*; *Laoukili et al., 2005*; *Wonsey and Follettie, 2005*), DNA repair (*Maachani et al., 2016*; *Park et al., 2012*; *Monteiro et al., 2013*; *Khongkow et al., 2014*; *Roh et al., 2020*), cell invasion and metastasis (*Wang et al., 2020*; *Luo et al., 2018*; *Mao et al., 2019*; *Parashar et al., 2020*), and chemoresistance (*Roh et al., 2020*; *Fang et al., 2018*; *Tassi et al., 2017*; *Kwok et al., 2010*; *Carr et al., 2010*; *Zhao et al., 2014*). Pan-cancer analyses have revealed that FOXM1 overexpression is widespread in human cancer and is linked to reduced patient survival and genomic instability (*Jiang et al., 2015*; *Li et al., 2017a*; *Barger et al., 2019*; *Gentles et al., 2015*; *Carter et al., 2006*). Accordingly, there is high interest in FOXM1 as a cancer therapeutic target (*Tabatabaei Dakhili et al., 2019*; *Halasi and Gartel, 2013b*; *Gormally et al., 2014*; *Ziegler et al., 2019*; *Xiang et al., 2017*).

High-grade serous ovarian cancer, also known as high-grade serous carcinoma (HGSC), is the most common and deadly subtype of epithelial ovarian cancer (EOC) (*Bowtell et al., 2015*). Molecular features of this tumor type include ubiquitous *TP53* mutations, defects in DNA homologous recombination repair (HR), *BRCA* gene mutations, and robust genomic instability characterized by widespread copy number alterations (CNAs) (*The Cancer Genome Atlas Research Network, 2011*). Notably, FOXM1 pathway activation is a characteristic of >85% of HGSC cases and is linked to

genomic instability (*Barger et al., 2019*; *The Cancer Genome Atlas Research Network, 2011*; *Barger et al., 2015*; *Li et al., 2020*). FOXM1 overexpression in HGSC and other cancers is driven by several mechanisms, including copy number gains, loss of function of p53 and pRB, activation of SP1, E2F1, and cyclin E1, and YAP signaling (*Barger et al., 2019*; *Barger et al., 2015*; *Fan et al., 2015*; *Petrovic et al., 2010*). In particular, *FOXM1* copy number gains in HGSC are localized to a region of chromosome 12p13.33 that contains 33 genes, and the *FOXM1* CNA is functional, meaning that *FOXM1* copy number directly correlates with *FOXM1* mRNA expression (*Barger et al., 2015*; *Song et al., 2019*). It is highly plausible that other genes in this amplicon contribute to HGSC, either independently or cooperatively with FOXM1.

The initial report of RHNO1 (first called C12orf32) showed that it was overexpressed in a breast cancer cell line, localized to the nucleus, and promoted cell survival (*Kim et al., 2010*). RHNO1 (reported as RHINO) was later identified in a high-throughput screen for DNA damage response (DDR) regulators and was shown to physically interact with the RAD9, RAD1, HUS1 DNA clamp (9-1-1). Through its interaction with 9-1-1 and TOPBP1, RHNO1 promotes the activation of ataxia telangiectasia and Rad3-related protein (ATR), a protein integral to the cellular DNA replication stress (RS) response (*Cotta-Ramusino et al., 2011*; *Lindsey-Boltz et al., 2015*). The RHNO1 N-terminus contains a conserved APSES DNA binding domain, necessary for interaction with the 9-1-1 complex, while the C-terminus is required for interaction with 9-1-1 and TOPBP1 and localization to sites of DNA damage (*Cotta-Ramusino et al., 2011*). Recently, X-ray crystallography was used to study the 9-1-1/RHNO1 interaction, and the data revealed that RHNO1 specifically interacts with RAD1 (*Hara et al., 2020*). The expression, regulation, and function of RHNO1 in normal tissues and cancer are largely unknown, but RHNO1 was recently identified as a hub gene associated with poor prognosis in colorectal cancer (*Yang et al., 2020*). The prominence of the RS response in cancer cell survival and its recent emergence as a therapeutic target make investigation of the function of RHNO1 in cancer an important area of study (*Ubhi and Brown, 2019*). Moreover, the ubiquity of elevated RS in HGSC makes this tumor type particularly relevant for RHNO1 investigation (*Konstantinopoulos et al., 2020*).

We report that *FOXM1* and *RHNO1* are bidirectional genes (BDGs) that are co-amplified in HGSC. We show that *FOXM1* and *RHNO1* are tightly co-expressed in both normal and cancer cells and tissues, including individual cells, and that their co-expression is controlled by a bidirectional promoter (F/R-BDP). We show that FOXM1 and RHNO1 both contribute to several important HGSC cell phenotypes, including clonogenic growth and cell survival, DNA HR repair, and PARP inhibitor (PARPi) resistance. Based on these data, we hypothesize that the FOXM1/RHNO1 BDG pair promotes HGSC and other cancers, and is a potential target for cancer therapeutic intervention.

## Results

### The 12p13.33 HGSC amplicon contains *FOXM1* and *RHNO1*, co-expressed BDGs

*FOXM1* is located at chromosome 12p13.33, a region showing frequent copy number gains in HGSC, and amplification at this locus is associated with reduced patient survival (*Barger et al., 2019*; *Barger et al., 2015*). Moreover, a recent study demonstrated that 12p13.33 amplification is detected from cell-free DNA in the plasma of HGSC patients at diagnosis and is further enriched in patients experiencing disease relapse (*Paracchini et al., 2020*). The minimal amplicon contains *FOXM1* and 32 additional genes (*Figure 1*). Based on recent studies of other cancer amplifications (*Justilien et al., 2014*; *Fields et al., 2016*), we hypothesized that additional genes at 12p13.33 might harbor oncogenic functions that act in concert with FOXM1. To test this hypothesis and search for gene candidates, we focused on genes showing correlated expression with *FOXM1* in HGSC. Using The Cancer Genome Atlas (TCGA) data, we found that, among all amplicon genes, *FOXM1* expression shows the strongest correlation with *RHNO1* (*Figure 2A*). Intriguingly, genomic analyses revealed that *FOXM1* and *RHNO1* are located next to each other and arranged 'head-to-head', that is, they are BDGs (*Figure 2B*). Greater than 10% of human genes are bidirectional, and the paired genes may be regulated by a bidirectional promoter (BDP) and have related functions (*Trinklein et al., 2004*; *Wakano et al., 2012*; *Yang et al., 2007*; *Tu et al., 2019*). Furthermore, cancer-related genes were recently shown to be highly enriched for BDG (*Chen et al., 2020*). Despite

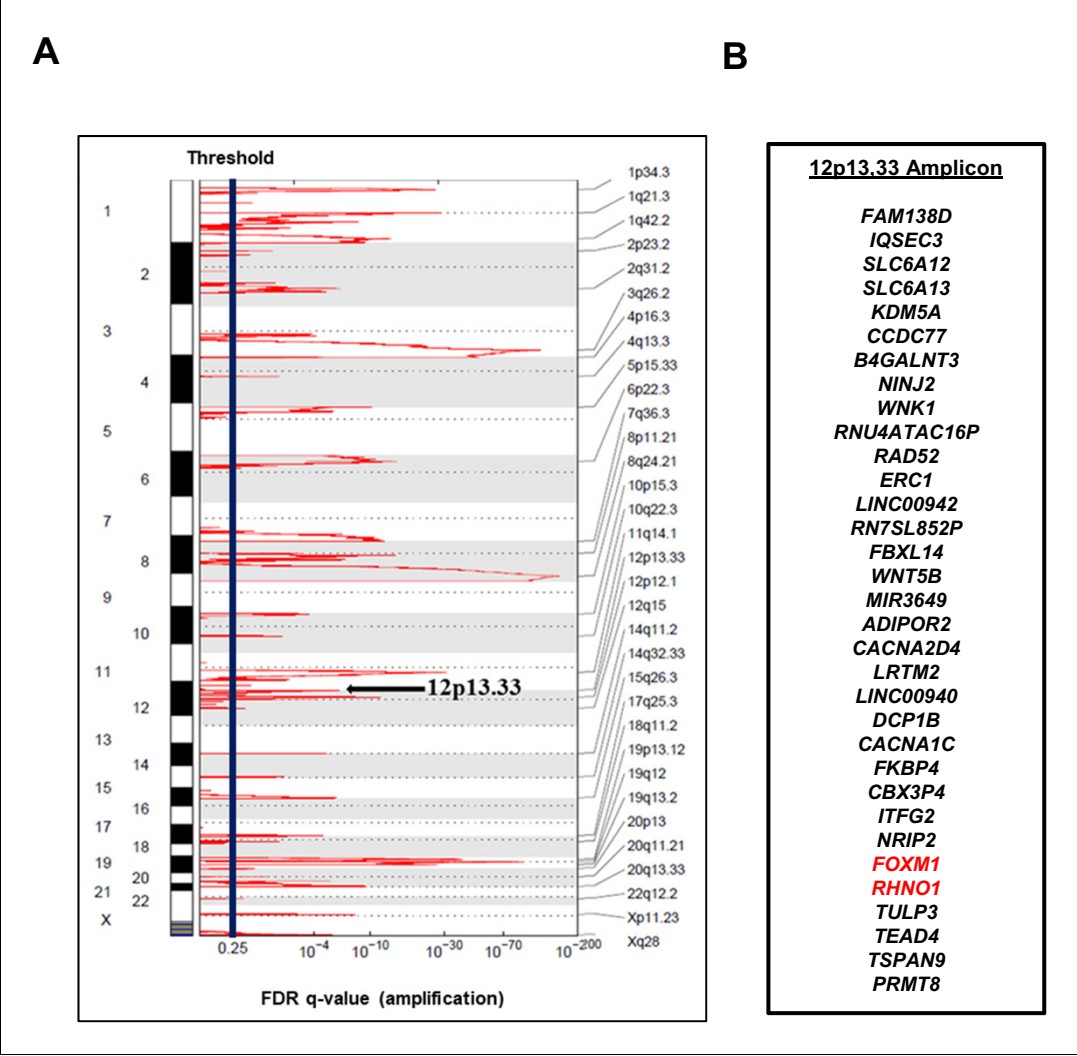

**Figure 1.** The 12p13.33 amplicon in high-grade serous carcinoma (HGSC). (**A**) Recurrent copy number amplifications in The Cancer Genome Atlas HGSC data (n = 579), as determined by GISTIC. Black vertical line indicates the threshold for significance, corresponding to an false discovery rate (FDR) q-value <0.25. The 12p13.33 genomic amplicon is indicated with an arrow. (**B**) The 33 genes in the 12p13.33 amplicon, listed top to bottom in genomic order from centromere to telomere. *FOXM1* and *RHNO1* are shown in red.

their strikingly high prevalence in the human genome and significant enrichment in cancer genes, the functional role of BDG has been infrequently studied (*Chen et al., 2020*; *Wang et al., 2015*; *Li et al., 2019*).

We next performed an in silico analysis of the putative *FOXM1/RHNO1* bidirectional promoter (F/R-BDP), that is, the genomic region spanning the putative *FOXM1/RHNO1* transcriptional start sites (TSS). The F/R-BDP contained a CpG island (CGI), a common feature of BDP (*Figure 2B*; *Wakano et al., 2012*; *Takai and Jones, 2004*; *Antequera, 2003*). Although differential DNA methylation of CGIs contained within BDP can regulate BDG expression (*Shu et al., 2006*), we found that the F/R-BDP is fully hypomethylated in both normal and cancer tissues, including normal ovary (NO) and ovarian cancer, suggesting that differential DNA methylation is not a regulatory mechanism at this BDP (*Figure 2—figure supplement 1*). In addition to a CGI, BDPs are often enriched with specific histone marks (*Bornelöv et al., 2015*), and The Encyclopedia of DNA Elements (ENCODE) data revealed the presence of bimodal peaks of transcriptionally active histone modifications at the F/R-BDP, strongly suggestive of bidirectional transcriptional activity (*Figure 2B*). Additionally, the F/R-BDP was characterized by DNase I hypersensitivity, enriched TF binding, active

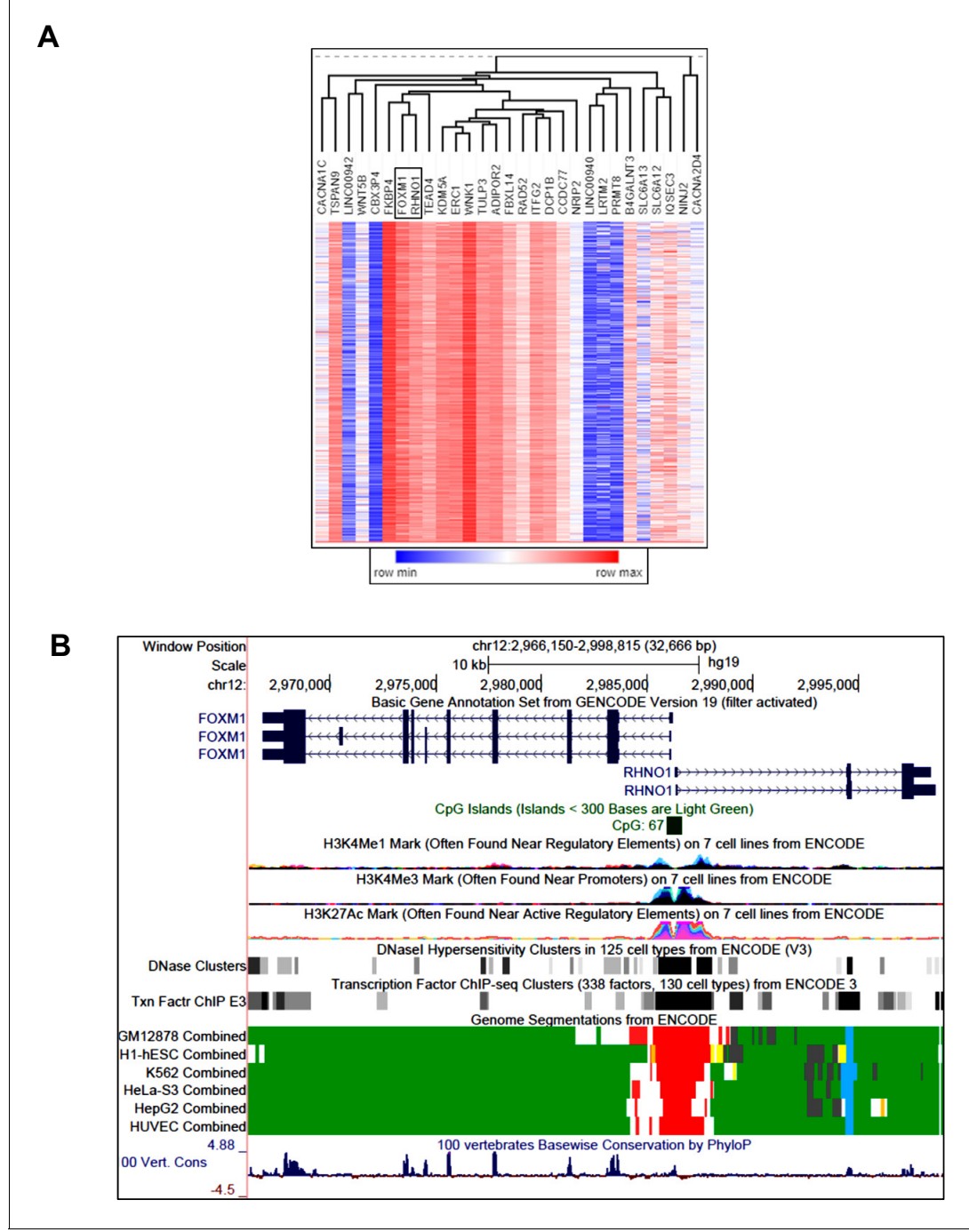

**Figure 2.** *FOXM1/RHNO1* expression correlation in high-grade serous carcinoma (HGSC) and genomic characteristics. (**A**) Hierarchical clustering dendrogram of The Cancer Genome Atlas HGSC mRNA expression correlations of the 33 genes in the 12p13.33 amplicon. *FOXM1* and *RHNO1* are boxed. (**B**) Genomic configuration and characteristics of *FOXM1* and *RHNO1*. Transcript structure, CpG islands, epigenetic marks, and vertebrate genome conservation data are shown. Data were obtained from the UCSC Genome Browser and ENCODE.

The online version of this article includes the following figure supplement(s) for figure 2:

**Figure supplement 1.** The *FOXM1/RHNO1* bidirectional promoter (F/R-BDP) region is hypomethylated in normal and cancer tissues.
**Figure supplement 2.** DNA sequence of the *FOXM1/RHNO1* bidirectional promoter (F/R-BDP).

chromatin segmentation, and vertebrate conservation (*Figure 2B*). Among TF binding sites in the F/R-BDP were E2F and MYC sites, factors that are known to regulate both BDPs and FOXM1 (*Barger et al., 2019*; *Pan et al., 2018*; *Bolognese et al., 2006*; *Figure 2—figure supplement 2*). Thus, the F/R-BDP displays several known characteristics of BDPs.

To determine whether copy number status associates not only with *FOXM1* but also *RHNO1* expression in HGSC, we analyzed TCGA and Cancer Cell Line Encyclopedia (CCLE) data (*The Cancer Genome Atlas Research Network, 2011*; *Barretina et al., 2012*). These analyses revealed a progressive increase in both *FOXM1* and *RHNO1* expression with increasing copy number (*Figure 3A, B*). We validated that FOXM1 and RHNO1 mRNA and protein expression correlated with copy number, using a laboratory collection of cell lines, including immortalized Fallopian tube epithelium (FTE) cells (a precursor cell type for HGSC) and HGSC cells (*Figure 3C, D*). Although the associations were significant, they were not uniform, indicating that mechanisms besides copy number status regulate *FOXM1* and *RHNO1* expression in HGSC. This is consistent with our prior studies of FOXM1 regulation (*Barger et al., 2019*; *Barger et al., 2015*).

## *FOXM1* and *RHNO1* expression correlates in bulk tissues, cell lines, and individual cells

We next analyzed whether *FOXM1* and *RHNO1* expression correlated in biological settings beyond HGSC. Consistent with TCGA HGSC data (*Figure 2A*), reverse transcription quantitative PCR (RT-qPCR) analyses of an independent set of primary EOC tissues, and a panel of ovarian surface

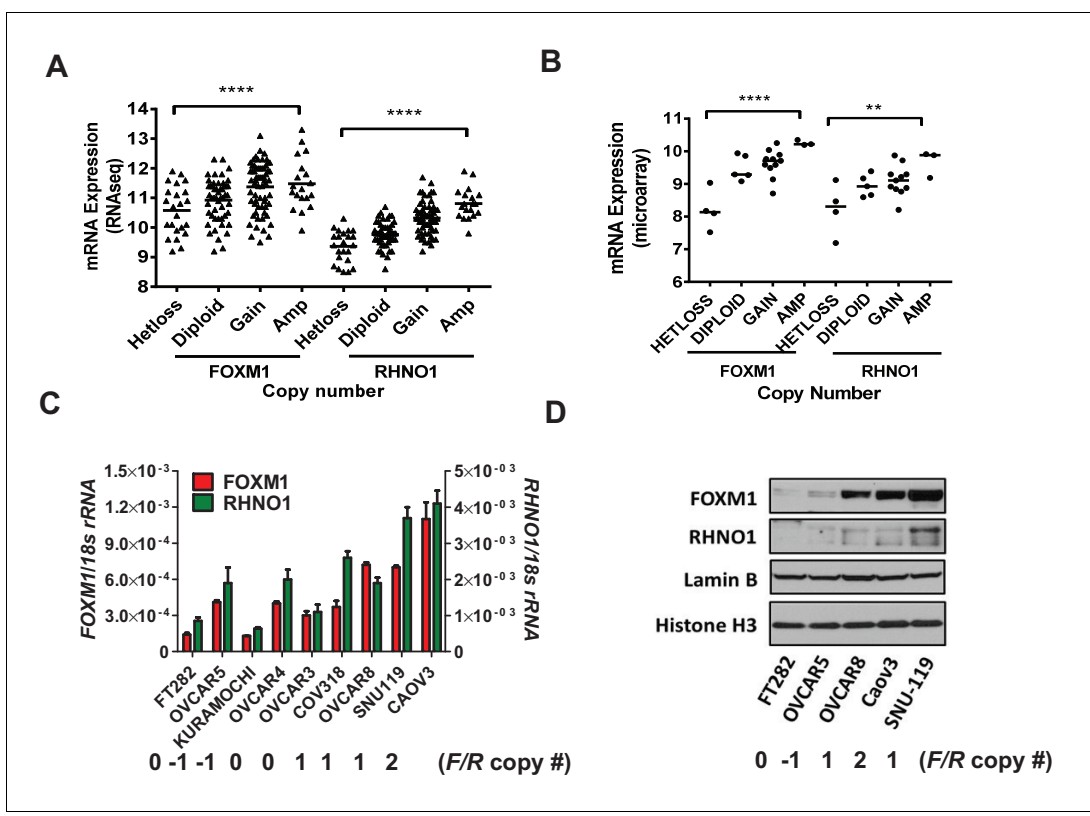

**Figure 3.** *FOXM1* and *RHNO1* expression correlates with genomic copy number in high-grade serous carcinoma (HGSC). (A) *FOXM1* and *RHNO1* mRNA expression (RNA-seq) vs. GISTIC copy number in The Cancer Genome Atlas HGSC data (N = 157). (B) *FOXM1* and *RHNO1* mRNA expression (microarray) vs. copy number (GISTIC) in Cancer Cell Line Encyclopedia HGSC cell lines (N = 23). p-values for ANOVA with post-test for linear trend are shown, and lines represent group medians. p-value designation: **p<0.01, ****p<0.0001. (C) *FOXM1* and *RHNO1* mRNA expression in FT282 immortalized Fallopian tube epithelium cells and HGSC cell lines (RT-qPCR). (D) FOXM1 and RHNO1 nuclear protein levels (western blot). Two nuclear protein loading controls are shown. *FOXM1/RHNO1* GISTIC copy number values for the cell lines are shown (−1: het loss; 0: diploid; 1: gain; 2: amp).

epithelium (OSE), FTE, and HGSC cell lines, indicated that *FOXM1* and *RHNO1* expression is highly correlated (*Figure 4A, B*). Next, to determine if this correlation occurs at the level of individual cells, we conducted single-cell RNA sequencing (scRNA-seq) of an immortalized human FTE cell line (FT282-C11, hereafter referred to as FT282) and a HGSC cell line (OVCAR8) (*Barger et al., 2018*;

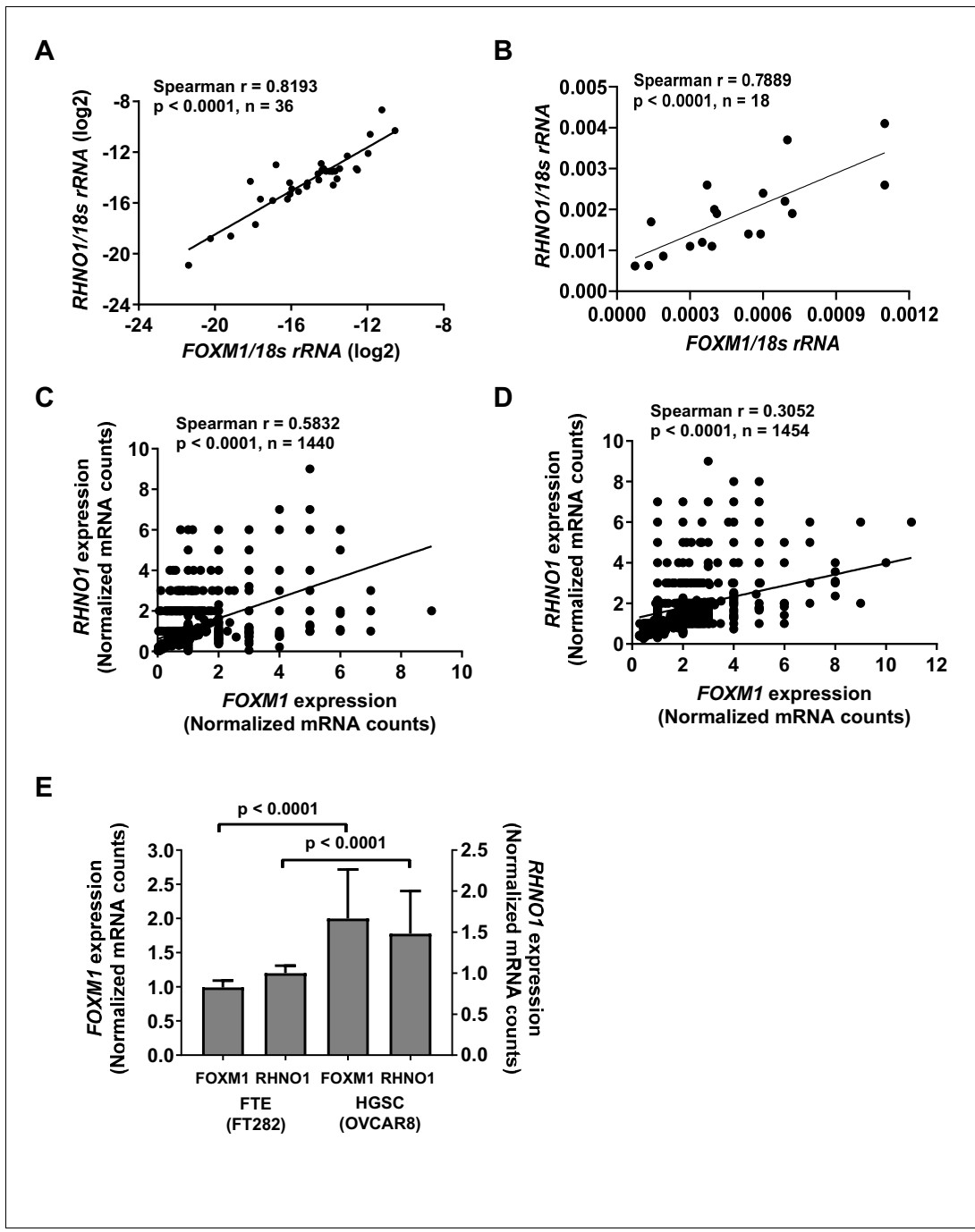

**Figure 4.** *FOXM1* and *RHNO1* expression correlates in high-grade serous carcinoma (HGSC) and in HGSC precursor cells. (A) *FOXM1* vs. *RHNO1* expression in primary HGSC tumors, as determined by RT-qPCR. (B) *FOXM1* vs. *RHNO1* expression in Fallopian tube epithelium (N = 5), ovarian surface epithelium (N = 3), and HGSC (N = 10) cell lines, as determined by RT-qPCR. (C) *FOXM1* vs. *RHNO1* expression in FT282 cells, as determined by scRNA-seq. (D) *FOXM1* vs. *RHNO1* expression in OVCAR8 HGSC cells, as determined by scRNA-seq. (A–D) show the Spearman r value, p-value, and the number of samples analyzed. (E) *FOXM1* and *RHNO1* expression in FT282 and OVCAR8 cells, plotting the median values from scRNA-seq data. Mann–Whitney p-values are shown.

*Domcke et al., 2013*). Importantly, *FOXM1* and *RHNO1* showed significant co-expression in each case (*Figure 4C, D*). Both genes showed elevated expression in OVCAR8 vs. FT282 cells, consistent with overexpression in cancer (*Figure 4E*). We next examined the degree of correlation seen between *FOXM1* and *RHNO1* as compared to other BDG genome-wide and, notably, observed that the *FOXM1/RHNO1* correlation ranked in the top 1% of all BDG in FT282 and in the top 9% in OVCAR8 cells (*Figure 5*). This observation supports a highly concordant regulatory process between *FOXM1* and *RHNO1* expression.

To address whether *FOXM1* and *RHNO1* expression correlates in biological contexts unrelated to ovarian cancer or its progenitor cells, we analyzed in silico gene expression data sets, including normal and cancer tissues and cells, and mouse and human data. Importantly, all comparisons revealed a highly significant correlation between *FOXM1* and *RHNO1*, indicating that co-expression is a defining feature of this BDG unit (*Supplementary file 1*).

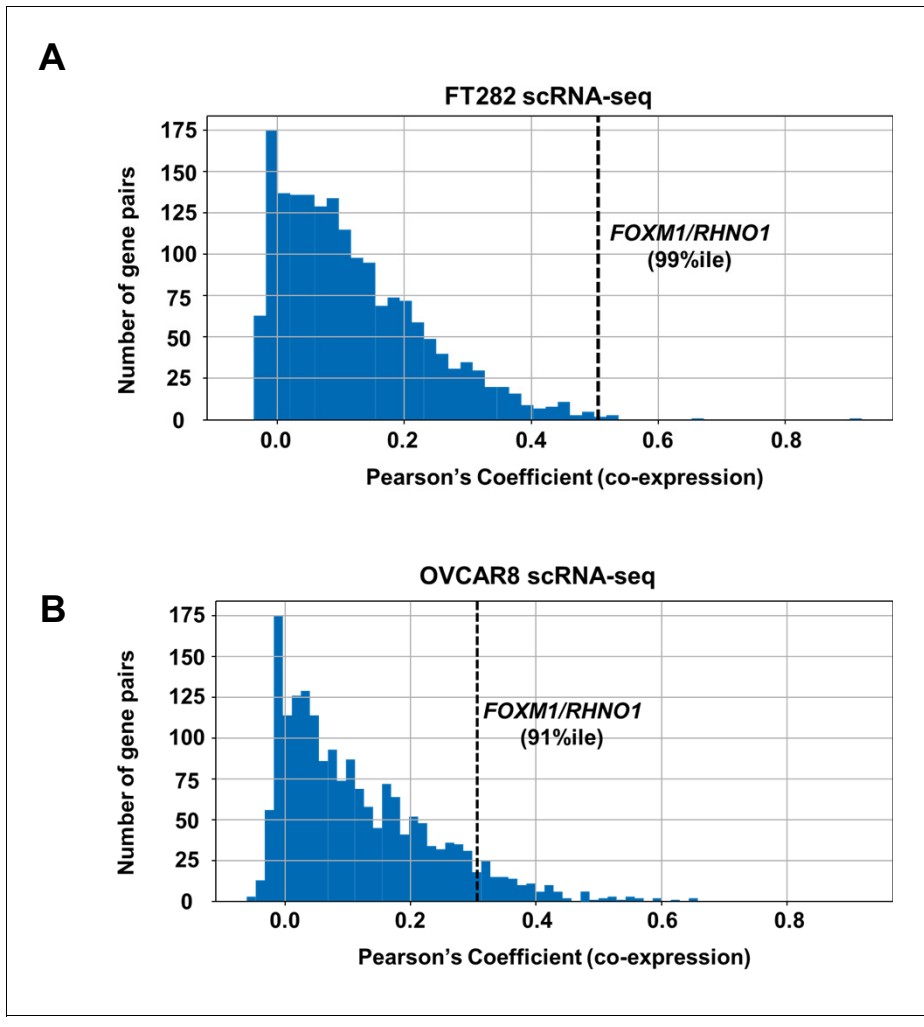

**Figure 5.** Distribution of correlation coefficients for bidirectional gene pairs (n = 2172) from scRNA-seq data. Data are shown for (**A**) FT282 and (**B**) OVCAR8 cell lines. The x-axis plots the degree of co-expression (Pearson's coefficient) between bidirectional gene pairs, and y-axis plots the frequency of bidirectional pairs. The rank percentile of the *FOXM1/RHNO1* correlation in each cell type is shown and is indicated by the dashed vertical line.

The online version of this article includes the following figure supplement(s) for figure 5:

**Figure supplement 1.** Principal component analyses (PCA) of FT282 (n = 1440) and OVCAR8 (n = 1454) single cells, using scRNA-seq data.

## FOXM1 and RHNO1 are overexpressed in pan-cancer

While *FOXM1* is overexpressed in HGSC and pan-cancer (*Barger et al., 2019*; *The Cancer Genome Atlas Research Network, 2011*; *Barger et al., 2015*), *RHNO1* expression in normal tissues vs. cancer is unknown. To address this question, we compared *FOXM1* and *RHNO1* expression in Genotype-Tissue Expression (GTEx) normal tissues vs. TCGA pan-cancer tissues using the Toil method (*Vivian et al., 2017*). Both *FOXM1* and *RHNO1* were overexpressed in each category of cancer samples (primary tumors, metastatic tumors, and recurrent tumors) as compared to normal tissues (*Figure 6A, B*). Moreover, *FOXM1* showed greater overexpression in tumor tissues compared to *RHNO1,* and, consistently, the *FOXM1/RHNO1* expression ratio was elevated in cancer (*Figure 6C*). This was in part driven by relatively higher *RHNO1* expression in normal tissues (*Figure 6B*). In addition, we hypothesized that increased *FOXM1/RHNO1* expression in tumors might reflect increased cell proliferation as *FOXM1* expression is closely associated with cell proliferation (*Ye et al., 1997*).

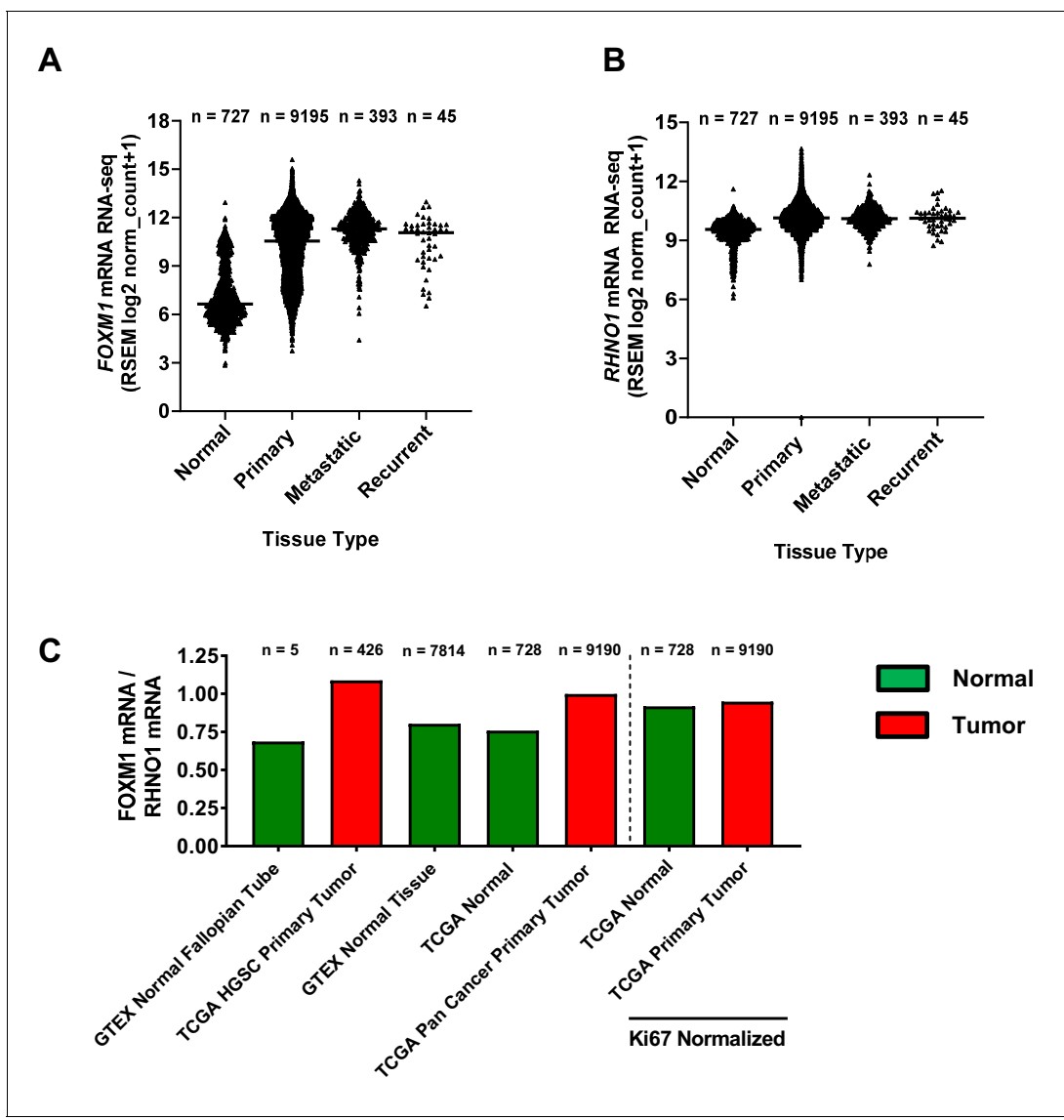

**Figure 6.** *FOXM1* and *RHNO1* mRNA expression and expression ratio in normal and cancer tissues. (A) *FOXM1* and (B) *RHNO1* expression (RNA-seq) in normal tissues (from Genotype-Tissue Expression and The Cancer Genome Atlas [TCGA]) vs. different types of TCGA tumor tissues (i.e., primary, metastatic, and recurrent). The number of samples in each group is shown. All comparisons of normal vs. tumor samples were highly significant (p<0.0001, t-tests). (C) *FOXM1/RHNO1* mRNA expression ratio in normal and cancer tissues, for the data shown in (A, B). The *FOXM1/RHNO1* ratio for the samples to the right of the dashed line was normalized to *MKI67* (a.k.a. Ki67), the canonical proliferation marker.

In agreement, after normalization to the canonical cell proliferation marker *MKI67* (*Whitfield et al., 2006*), the *FOXM1/RHNO1* expression ratio was similar in normal and cancer tissues (*Figure 6C*).

## *FOXM1* and *RHNO1* expression in HGSC

Next, we examined the expression of *FOXM1* and *RHNO1* in HGSC. HGSC has traditionally been proposed to emanate from the OSE, but is now believed to predominantly, but not exclusively, arise from the FTE (*Soong et al., 2019*). Comparison of TCGA HGSC vs. GTEx Fallopian tube RNA-seq data revealed that both *FOXM1* and *RHNO1* expression show relatively uniform and highly significant increases in HGSC vs. Fallopian tube (*Figure 7A*). In agreement, *FOXM1* and *RHNO1* were both significantly elevated in HGSC cell lines as compared to primary or immortalized FTE/OSE cells (*Figure 7B*).

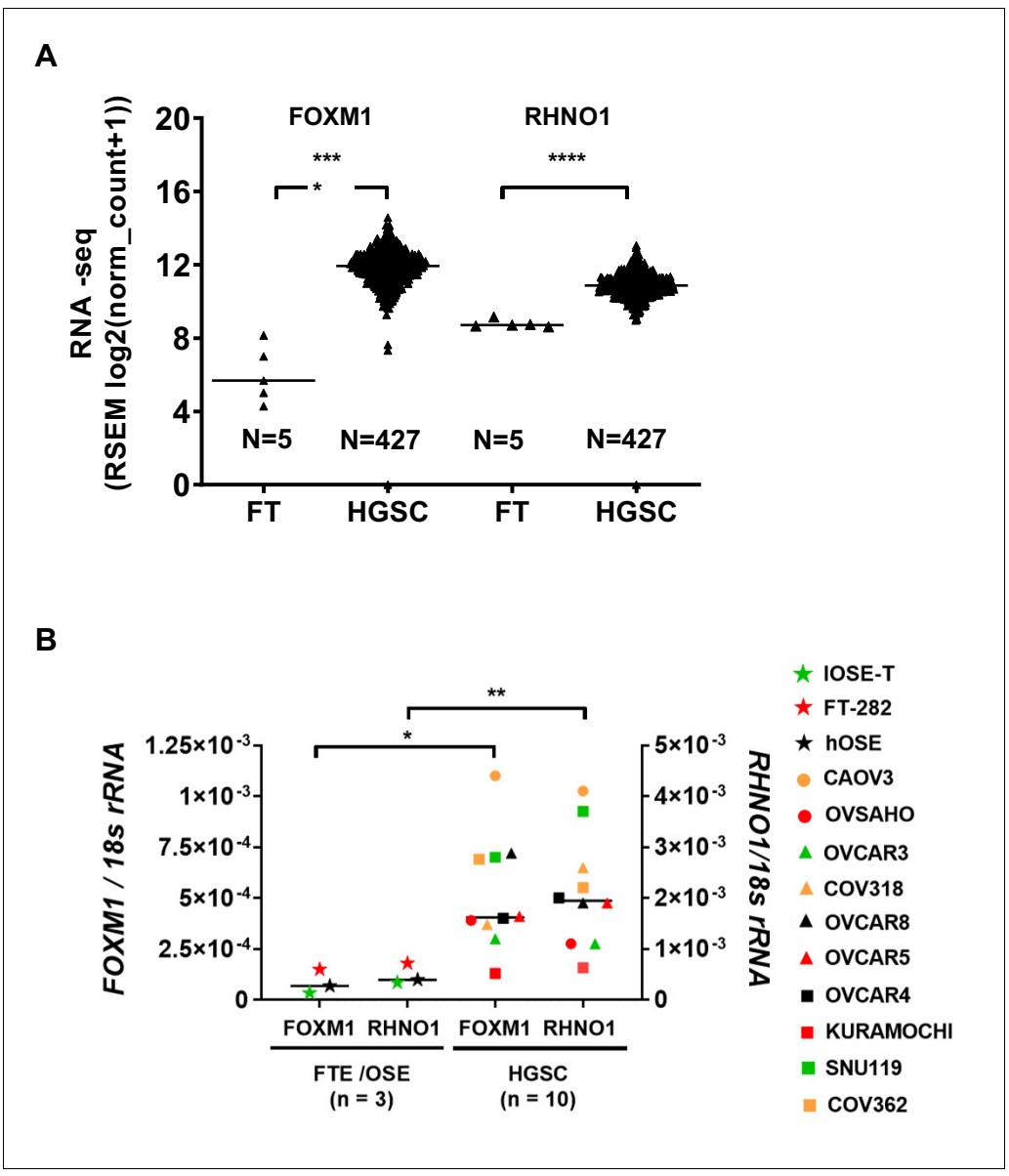

**Figure 7.** *FOXM1* and *RHNO1* are overexpressed in high-grade serous carcinoma (HGSC). (**A**) *FOXM1* and *RHNO1* expression in Genotype-Tissue Expression normal Fallopian tube (FT; N = 5) vs. The Cancer Genome Atlas HGSC tissues (RNA-seq) (N = 427). (**B**) *FOXM1* and *RHNO1* mRNA expression in Fallopian tube epithelium/ovarian surface epithelium cells vs. HGSC cell lines (RT-qPCR). t-test p-values are shown. p-value designation: ****<0.0001, ***<0.001, **<0.01, *<0.05. The key shown at right indicates the cell types for each data point.

Following front-line therapy, most clinically advanced HGSC patients relapse with a chemoresistant recurrent disease, so developing improved treatments for these patients remains critical (*Bowtell et al., 2015*). We compared *FOXM1* vs. *RHNO1* expression in primary and recurrent HGSC samples using two independent patient cohorts (*Patch et al., 2015*; *Kreuzinger et al., 2017*). In both cohorts, *FOXM1* and *RHNO1* expression significantly correlated in both primary and recurrent disease (*Figure 8A, B*). While, overall, *FOXM1* and *RHNO1* expression wa s not significantly altered in primary vs. recurrent disease, 5/11 patients displayed coordinated upregulation of both *FOXM1* and *RHNO1* at recurrence (*Figure 8C*). This observation suggests that *FOXM1* and *RHNO1* might contribute to disease recurrence in a large subset of HGSC patients.

## *FOXM1* and *RHNO1* expression is co-regulated by a BDP

We used functional assays to assess whether *FOXM1* and *RHNO1* expression is co-regulated by the F/R-BDP. Initially, we determined the TSS of *FOXM1* and *RHNO1* in FTE (FT282) and HGSC (OVCAR4, OVCAR8) cells using 5′ RNA ligase-mediated rapid amplification of cDNA ends (RLM-RACE). The data revealed TSSs consistent with NCBI Genome Browser data, revealing an intergenic span (i.e., between the 5′ ends of *FOXM1* and *RHNO1*) of approximately 150 bp (*Figure 9—figure supplement 1*). These data also show that the TSS are consistent between normal vs. HGSC cells, and confirm that the TSS is <1 kb, which is characteristic of BDPs. To define transcriptional activity at the F/R-BDP, we generated a luciferase reporter construct consisting of a 190 bp segment that includes the intergenic region as well as small portions of the 5′ ends of *FOXM1* and *RHNO1*. In this construct, the *FOXM1* promoter direction drives Renilla luciferase while the *RHNO1* promoter direction drives Firefly luciferase, from the opposite DNA strand (*Figure 9A*). Studies in FTE and HGSC

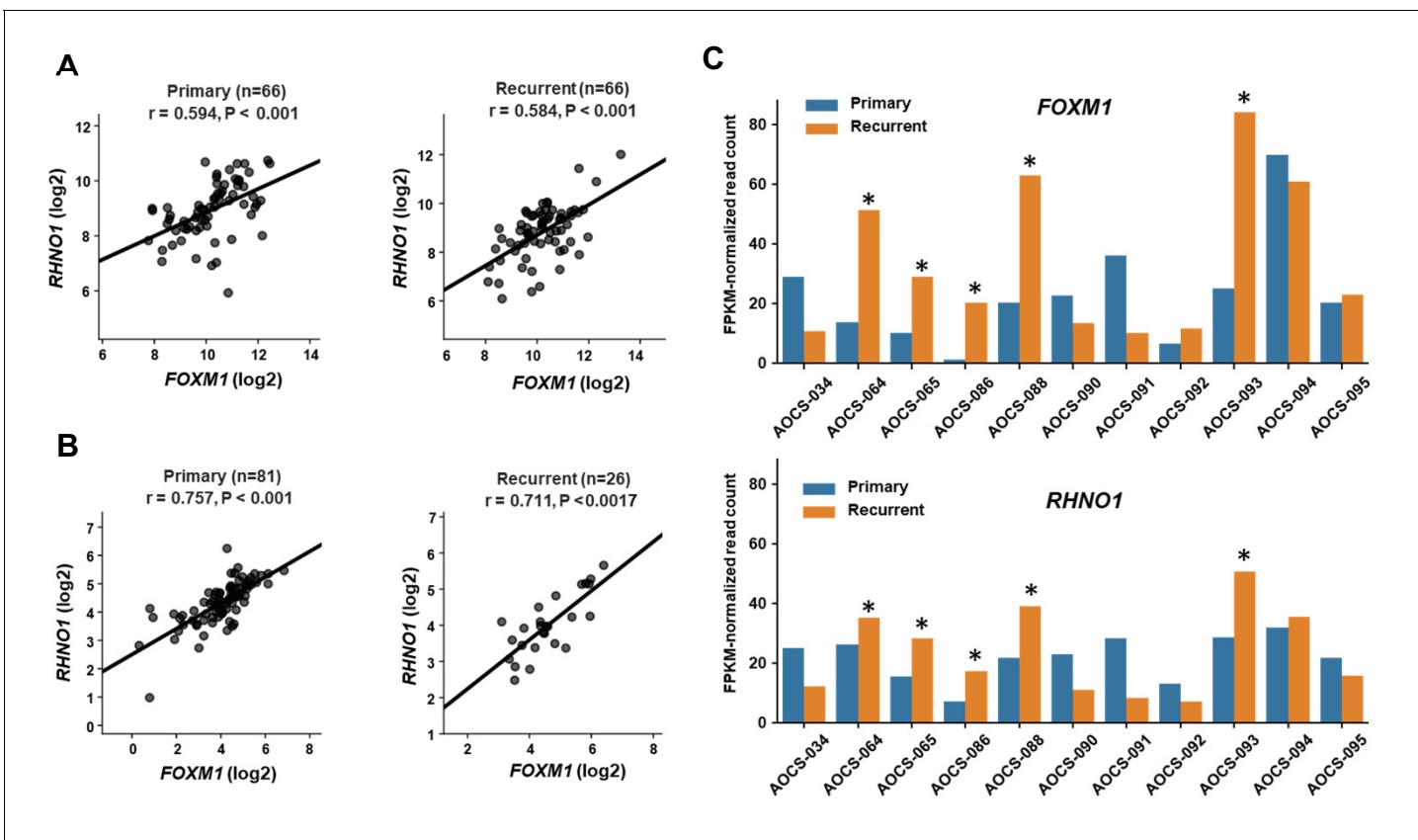

**Figure 8.** *FOXM1* and *RHNO1* expression in primary and recurrent high-grade serous carcinoma (HGSC). (**A**) *FOXM1* vs. *RHNO1* expression (RNA-seq data) in primary (left) and recurrent (right) HGSC samples from *Kreuzinger et al., 2017*. (**B**) *FOXM1* vs. *RHNO1* expression (RNA-seq data) in primary (left) and recurrent (right) HGSC samples from *Patch et al., 2015*. For (**A**, **B**) , the number of samples (n), Spearman r value, and p-value are shown. (**C**) *FOXM1* (top) and *RHNO1* (bottom) expression (RNA-seq) in patient-matched primary vs. recurrent HGSC samples from *Patch et al., 2015*. Asterisks indicate patients showing coordinately increased expression of both *FOXM1* and *RHNO1* in recurrent tumors (5/11 = 45% of patients).

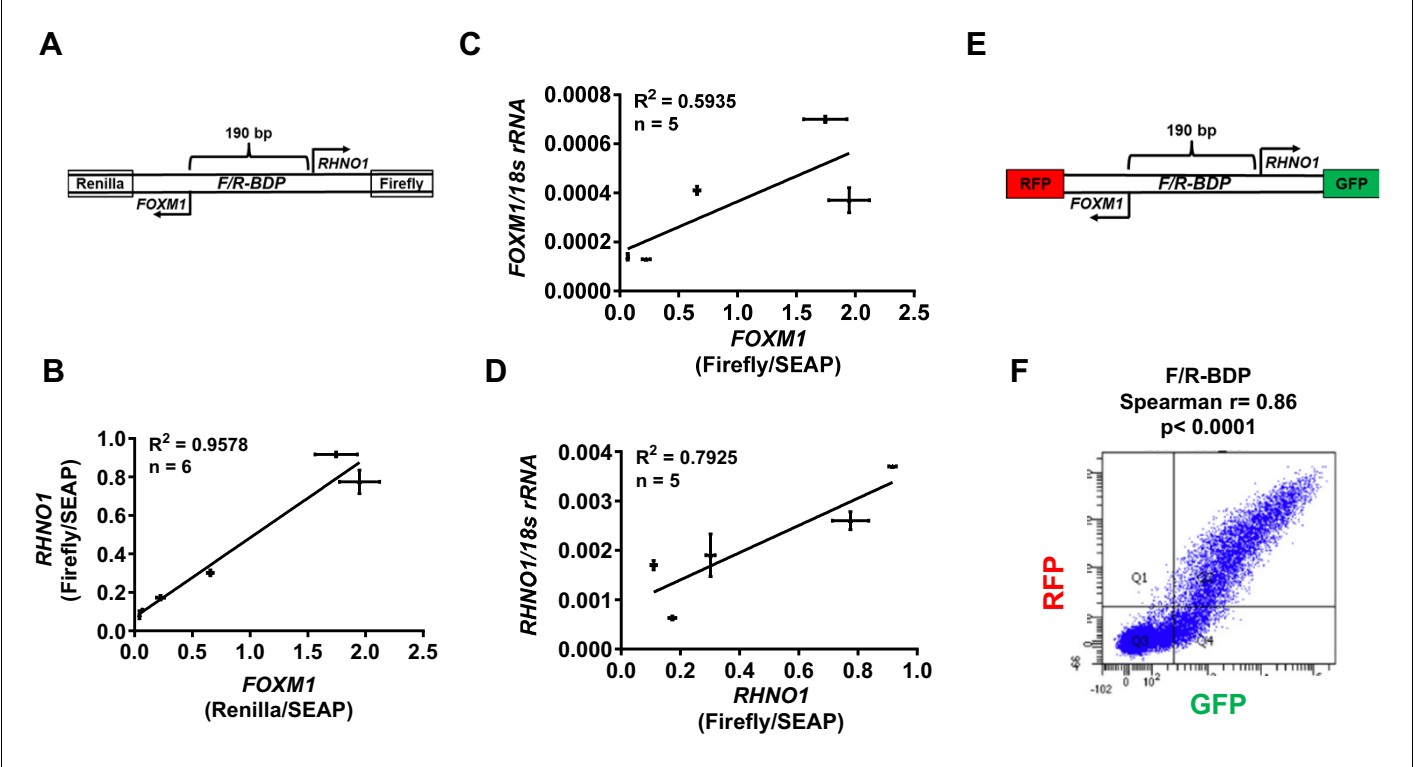

**Figure 9.** F/R-BDP promoter activity assays. (**A**) Schematic of the *F/R-BDP* dual luciferase reporter construct. The *FOXM1* promoter direction drives Renilla luciferase, and the *RHNO1* promoter direction drives firefly luciferase. Secreted embryonic alkaline phosphatase (SEAP) was used as a normalization control for transfection efficiency. (**B**) Correlation of *FOXM1* and *RHNO1* promoter activity in Fallopian tube epithelium (FTE) (FT282) and high-grade serous carcinoma (HGSC) cell lines (OVCAR5, COV318, OVCAR4, Kuramochi, SNU119). (**C, D**) Correlation of *FOXM1* or *RHNO1* promoter activity vs. endogenous expression of the cognate gene in FTE (FT282) and HGSC (OVCAR5, COV318, Kuramochi, SNU119) cell lines, as measured by RT-qPCR. In (**B–D**), error bars represent standard deviations. (**E**) Schematic of the *F/R-BDP* dual fluorescence reporter construct. The *FOXM1* promoter direction drives red fluorescent protein (RFP) expression, and the *RHNO1* promoter direction drives green fluorescent protein (GFP) expression. (**F**) Fluorescence-activated cell sorting analysis of GFP and RFP fluorescence in 293T cells transfected with the F/R-BDP reporter shown in (**E**). Quadrant 2 (RFP + GFP + cell population) correlation statistics are shown.

The online version of this article includes the following figure supplement(s) for figure 9:

**Figure supplement 1.** *FOXM1* and *RHNO1* transcriptional start sites.

**Figure supplement 2.** Fluorescence-activated cell sorting analyses of red fluorescent protein (RFP) and green fluorescent protein (GFP) expression in 293T cells.

cells using this construct verified BDP activity and also revealed a direct correlation between *FOXM1* and *RHNO1* promoter activity (***Figure 9B***). Moreover, *FOXM1* and *RHNO1* promoter activity significantly correlated with expression of the cognate endogenous mRNA (***Figure 9C, D***). This correlation was not perfect, suggesting that additional mechanisms, such as enhancer activity or post-transcriptional processes, regulate expression of the endogenous genes. We used a complementary approach to assess F/R-BDP activity in individual cells, wherein we generated a reporter construct where the *FOXM1* promoter direction drives green fluorescent protein (GFP) and the *RHNO1* promoter direction drives red fluorescent protein (RFP) expression, which can be measured by flow cytometry (***Figure 9E***). Experiments using this construct demonstrated a significant direct correlation between *FOXM1* and *RHNO1* promoter activity in individual cells (***Figure 9F***, ***Figure 9—figure supplement 2***). This observation is consistent with the results of our scRNA-seq experiment that measured endogenous *FOXM1* and *RHNO1* expression in individual cells (***Figure 4C, D***).

Although promoter reporter assays are informative, they do not fully reflect mechanisms regulating endogenous genes. To address this, we used CRISPR-dCas9 and single-guide RNA (sgRNA) to deliver activator (VP64, i.e., CRISPRa) or repressor (KRAB, i.e., CRISPRi) proteins to the F/R-BDP region (***Joung et al., 2017***; ***Thakore et al., 2015***). We used three sgRNA to directly target either

the F/R-BDP or flanking regions (*Figure 10A*). We first used CRISPRa in FT282 cells, which have low levels of endogenous *FOXM1* and *RHNO1* expression. sgRNA targeting outside the putative BDP, upstream of one gene and in the gene body of the other gene (i.e., sg070 or sg233), induced the expression of the distal (i.e., upstream) gene (*Figure 10B*). In contrast, sgRNA directly targeting the F/R-BDP (i.e., sg130) induced the expression of both *FOXM1* and *RHNO1*. Next, we used CRISPRi in OVCAR8 cells, which have high endogenous expression of *FOXM1* and *RHNO1*. CRISPRi targeting the F/R-BDP (sg130) efficiently repressed both *FOXM1* and *RHNO1*, while guides flanking the F/R-BDP had a lesser effect (*Figure 10C*). Also, sg233, located in *RHNO1* exon 1, had a greater repressive effect on both genes compared to sg070, which is located in *FOXM1* exon 1. One potential explanation for this result is that endogenous transcriptional activators of both genes have increased localization to *RHNO1* exon 1 compared to *FOXM1* exon 1. To validate these observations, we

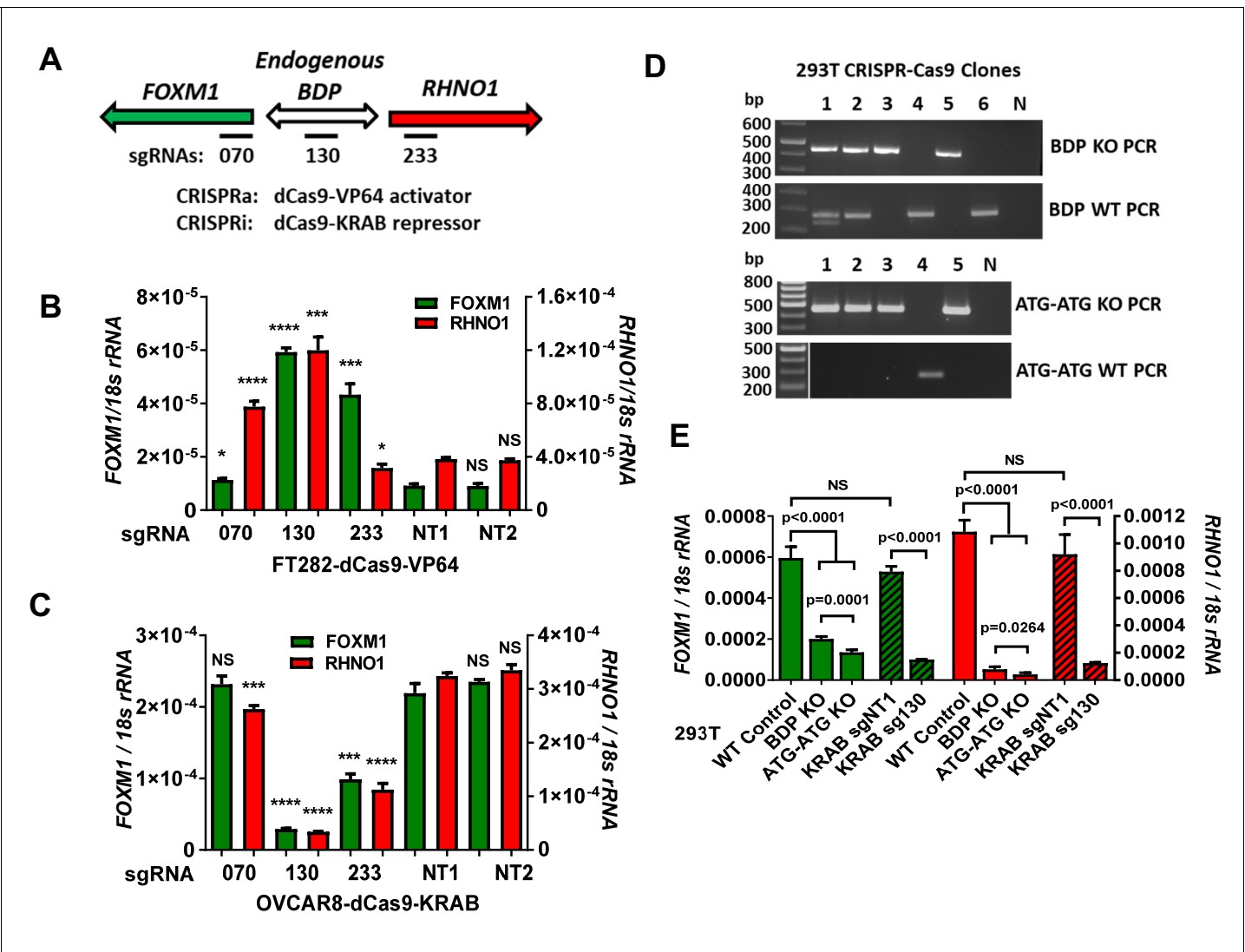

**Figure 10.** Endogenous *FOXM1* and *RHNO1* are co-regulated by their bidirectional promoter (F/R-BDP). (A) Schematic of the experimental approach used in (B, C). The position of the CRISPR guide RNAs is indicated with black rectangles. (B) CRISPRa experiment in FT282 cells. Endogenous *FOXM1* and *RHNO1* expression was measured by RT-qPCR. (C) CRISPRi experiment in OVCAR8 cells. Endogenous *FOXM1* and *RHNO1* mRNA expression was measured by RT-qPCR. In (B, C), NT1 and NT2 indicate non-targeting control guide RNAs, and t-test p-values are shown for the comparison with NT1 control for either *FOXM1* or *RHNO1*. p-values: ****<0.0001, ***<0.001, **<0.01, *<0.05. (D) Genotyping data for 293T cells with knockout (KO) of the 200 bp F/R-BDP (BDP KO) or, as a control, the entire upstream regions (5' of the translational start sites) of both *FOXM1* and *RHNO1* (ATG-ATG KO). N: no template control. (E) Endogenous *FOXM1* and *RHNO1* expression in 293T cells with the indicated genetic configurations. BDP KO and ATG-ATG KO correspond to cells described in (D). KRAB data for 293T cells (i.e., CRISPRi), using the strategy shown in (A), are shown for comparison on the right.

performed CRISPR knockout and CRISPRi experiments in 293T cells, which have high levels of endogenous *FOXM1* and *RHNO1* expression. These experiments further validated that the endogenous F/R-BDP co-regulates *FOXM1* and *RHNO1* (*Figure 10D, E*).

## FOXM1 and RHNO1 promote HGSC cell clonogenic growth

Based on their genomic configuration, co-amplification, co-expression, and overexpression in HGSC, we hypothesized that FOXM1 and RHNO1 may each promote HGSC oncogenic phenotypes. Moreover, we noted that BDGs are enriched as cancer genes and in specific functions, including cell cycle and DNA repair, and may function cooperatively (*Chen et al., 2020*; *Wang et al., 2015*; *Li et al., 2019*; *Li et al., 2006*; *Sun et al., 2021*). We thus focused our investigations on known and potentially overlapping functions of FOXM1 and RHNO1, including clonogenic cell growth and survival, cell cycle, and DNA repair (*Barger et al., 2015*; *Kim et al., 2010*; *Cotta-Ramusino et al., 2011*; *Lindsey-Boltz et al., 2015*; *Zona et al., 2014*). We utilized two validated HGSC cell lines (OVCAR8 and CAOV3) for these studies (*Domcke et al., 2013*; *Haley et al., 2016*). These two lines express both FOXM1 and RHNO1 and harbor copy number gains at 12p13.33 (*Figure 3C, D*). We initially engineered the cell lines for doxycycline (dox)-inducible lentiviral expression of shRNA targeting either FOXM1 or RHNO1. We used this knockdown approach due to reported off-target effects of CRISRP-Cas9 gene targeting at amplified loci (*Aguirre et al., 2016*). The engineered lines demonstrated efficient gene knockdown 72 hr after dox treatment (*Figure 11A*). Notably, either FOXM1 or RHNO1 knockdown reduced OVCAR8 and CAOV3 clonogenic growth (*Figure 11B*, *Figure 11—figure supplement 1*). To validate this observation (despite potential concerns with CRISPR-Cas9 targeting), we engineered OVCAR8 and CAOV3 cells for polyclonal CRISPR knockout of *FOXM1* or *RHNO1* (*Figure 11C*). The results verified that disruption of either gene reduces clonogenic growth of both HGSC cell lines (*Figure 11D*, *Figure 11—figure supplements 1 and 2*). Further experiments indicated that clonogenic growth disruption in OVCAR8 CRISPR knockout cells is caused in part by increased apoptosis (*Figure 11—figure supplement 1*).

To determine the combined effect of FOXM1 and RHNO1 depletion, we engineered OVCAR8 HGSC cells for dual FOXM1 and RHNO1 shRNA knockdown (*Figure 12A*). Dual knockdown reduced OVCAR8 clonogenic growth to a greater degree than either single knockdown, although the effect was similar to that observed with RHNO1 knockdown alone (*Figure 12B, C*). To confirm these observations, we constructed CAOV3 HGSC cells with dual FOXM1 + RHNO1 knockdown. The data again showed greatest loss of clonogenic growth with dual FOXM1/RHNO1 knockdown (*Figure 12—figure supplement 1*).

## RHNO1 promotes ATR-Chk1 signaling in HGSC cells

RHNO1 interacts with the 9-1-1 complex and TOPBP1 to promote ATR activation (*Cotta-Ramusino et al., 2011*; *Lindsey-Boltz et al., 2015*), but this has not been assessed in cancer cells or after treatment with specific inducers of RS To address these questions, we first ectopically expressed HA-tagged RHNO1-WT (wild-type) and the RHNO1-SWV mutant in 293T cells and performed Co-IP/western blotting. RHNO1-SWV has mutations in the APSES DNA binding domain that disrupt its interaction with 9-1-1 but not TOPBP1 (*Figure 13A*; *Cotta-Ramusino et al., 2011*). As expected, RHNO1-WT interacted with RAD9 and RAD1 in 293T cells while RHNO1-SWV was deficient for these interactions (*Figure 13B*). Next, we assessed RHNO1 functions in OVCAR8 HGSC cells. RHNO1 was present in chromatin protein extracts at baseline, while treatment with hydroxyurea (HU), a specific inducer of RS, increased its chromatin localization (*Figure 13C*). To test RHNO1 interaction with the 9-1-1 complex, we depleted endogenous RHNO1 from OVCAR8 cells using dox-inducible shRNA knockdown, with concomitant dox-inducible expression of shRNA-resistant HA-tagged RHNO1-WT or RHNO1-SWV. The data showed that RHNO1-WT interacted with 9-1-1 and TOPBP1, but RHNO1-SWV interacted only with TOPBP1 (*Figure 13D*). Next, to assess the function of RHNO1 in ATR-Chk1 signaling in HGSC cells, we depleted RHNO1 from OVCAR8 cells using dox-inducible shRNA and measured P-Chk1-S345, the canonical readout of the cellular response to RS (*Niida et al., 2007*). We observed an approximately twofold reduction of P-Chk1-S345 in RHNO1-depleted OVCAR8, a similar effect to that reported previously (*Cotta-Ramusino et al., 2011*; *Lindsey-Boltz et al., 2015*; *Figure 13E*). To test RHNO1 function in DNA damage protection, we measured a canonical marker of DNA breaks (γ-H2AX) in different cell cycle phases in OVCAR8

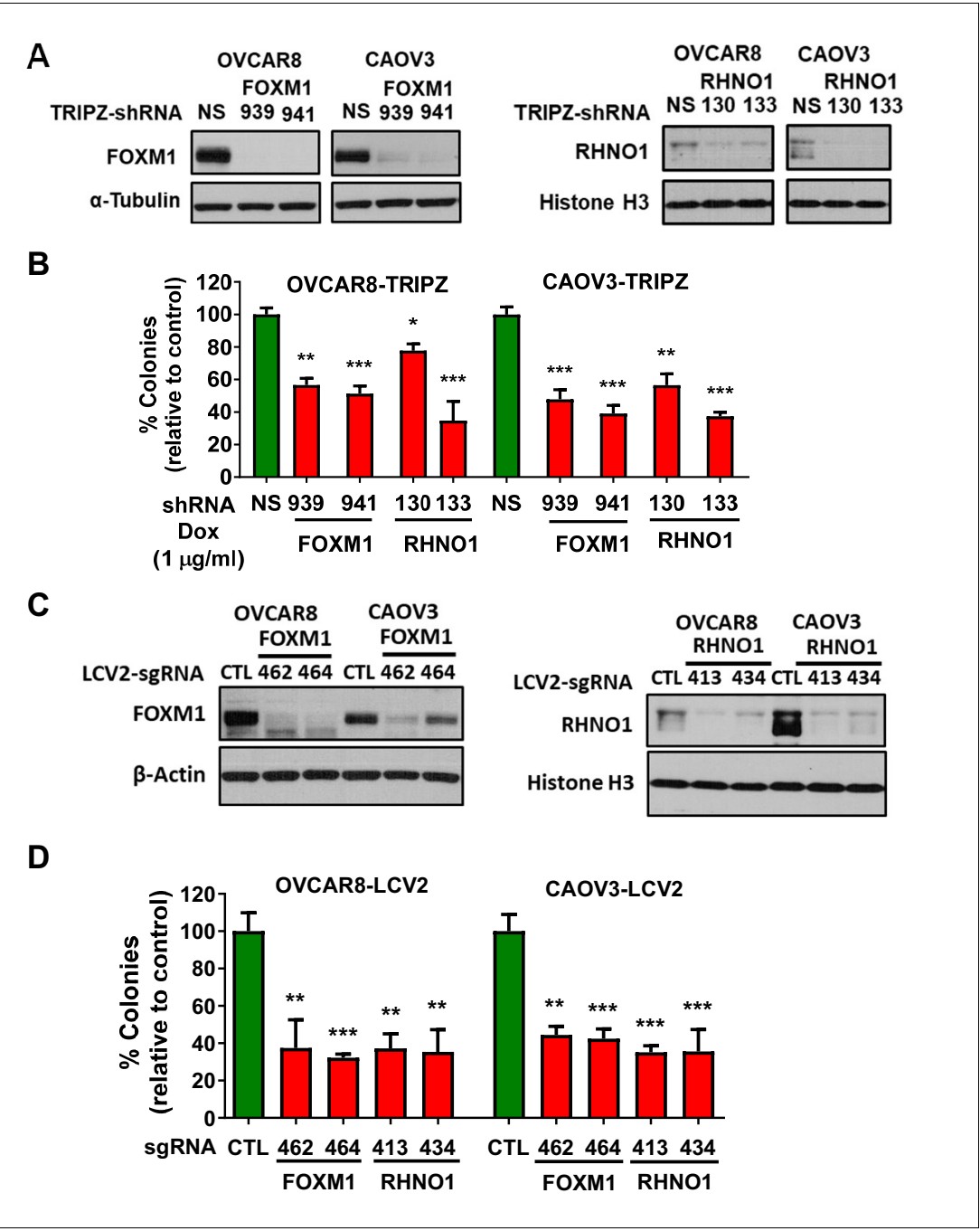

**Figure 11.** FOXM1 and RHNO1 each promote high-grade serous carcinoma cell clonogenic growth. (**A**) OVCAR8 and CAOV3 cells engineered for doxycycline (dox)-inducible FOXM1 or RHNO1 shRNA knockdown. Cells were grown in the presence of dox for 72 hr, and protein was extracted for western blot analyses. (**B**) Cells were seeded into 6-well dishes, in triplicate. Dox was added at the time of seeding, and media containing dox was replenished every 48 hr. Clonogenic growth was measured at 12–14 days, and the number of colonies (>50 cells) was counted. NS: nontargeting shRNA. (**C**) OVCAR8 and CAOV3 cells were engineered for FOXM1 or RHNO1 CRISPR knockout. Western blot analysis of polyclonal cell populations was performed 72 hr post-lentivirus infection. CTL: control guide RNA. (**D**) Clonogenic growth was measured as described in (**B**). t-test p-values vs. control: ****<0.0001, ***<0.001, **<0.01, *<0.05.

The online version of this article includes the following figure supplement(s) for figure 11:

**Figure supplement 1.** FOXM1 and RHNO1 each promote high-grade serous carcinoma cell clonogenic growth.

*Figure 11 continued on next page*

*Figure 11 continued*

**Figure supplement 2.** Representative TIDE analyses data for high-grade serous carcinoma cells sustaining FOXM1 or RHNO1 CRISPR gene knockout.

cells. We observed a significant increase of γ-H2AX- in both S and G2 phase after RHNO1 knockdown (*Figure 13F*). Additionally, COMET analyses revealed increased DNA strand breaks in OVCAR8 cells after RHNO1 knockdown (*Figure 13G*). These data establish that RHNO1 functions in the RS response and DNA damage protection in HGSC cells.

## RHNO1 binding to 9-1-1 is required for its role in the promotion of HGSC clonogenic growth

To determine whether RHNO1 interaction with 9-1-1 is required for promotion of clonogenic growth, we knocked down endogenous RHNO1 using dox-inducible shRNA in OVCAR8 cells and simultaneously ectopically expressed dox-inducible, shRNA-resistant, RHNO1-WT or RHNO1-SWV (*Figure 14A, B*). Importantly, RHNO1-SWV did not rescue the clonogenic growth defect while RHNO1-WT partially rescued the phenotype (*Figure 14C, D*). These data link RHNO1 function in clonogenic growth to its role in the RS response via binding to the 9-1-1 complex.

## FTE cells with RHNO1 knockout are viable and proliferate normally

The cell growth and viability studies described above were performed using HGSC cell lines, and the only prior publication linking RHNO1 to this phenotype utilized a breast cancer cell line (*Kim et al., 2010*). To examine whether RHNO1 also contributes to the growth and survival of normal (nontransformed) cells, we used CRISPR-Cas9 to eliminate *RHNO1* in immortalized FTE cells (FT282), a model of HGSC precursor cells (*Karst et al., 2014*; *Figure 15A, B*). Remarkably, FTE cells survived longterm expansion upon RHNO1 homozygous knockout and did not exhibit altered growth kinetics (*Figure 15C*). Clonogenic assays are not possible in this cell type as these cells do not form colonies (data not shown). This finding suggests that cancer cells may have an increased dependency on RHNO1 compared to nontransformed cells, possibly due to elevated RS in cancer cells (*Buisson et al., 2015*). To test this, we compared the expression of two established markers of RS (P-CHK1-Ser345 and P-RPA2-Ser33) in FT282 cells and HGSC cell lines. The data reveal that HGSC cell lines, including OVCAR8, have robust elevation in RS biomarkers compared to FT282 cells (*Figure 15D*). Together, these data, along with the large elevation of RHNO1 expression seen in cancer vs. normal (*Figures 6* and *7*), suggest that cancer cells have a greater dependency on RHNO1 than do normal cells.

## FOXM1 and RHNO1 promote DNA homologous recombination repair (HR)

Based on prior studies of FOXM1 and RHNO1 (*Cotta-Ramusino et al., 2011*; *Maachani et al., 2016*; *Park et al., 2012*; *Monteiro et al., 2013*; *Khongkow et al., 2014*), we investigated the function of these genes in promoting HR. For this task, we initially used U2OS cells engineered with a functional DNA repair reporter assay (DR-GFP), a widely used model for quantifying HR proficiency in single cells (*Gunn et al., 2011*; *Gunn and Stark, 2012*). We further engineered U2OS DR-GFP cells for dox-inducible FOXM1 and/or RHNO1 shRNA knockdown, and also engineered RAD51 knockdown as a positive control for HR impairment (*Figure 16A, B*; *Cotta-Ramusino et al., 2011*). The data revealed that RHNO1, but not FOXM1, knockdown significantly decreased HR capacity in U2OS cells. Interestingly, dual knockdown of FOXM1 and RHNO1 resulted in greater HR impairment, suggesting a cooperative interaction between FOXM1 and RHNO1 in promoting HR in this cell type (*Figure 16C*). Next, to address HR in the context of HGSC cells, we used OVCAR8 DR-GFP cells. OVCAR8 cells have monoallelic *BRCA1* promoter methylation but retain HR proficiency, similar to other HGSC cell lines with *BRCA1* methylation (*Huntoon et al., 2013*; *Karakashev et al., 2020*). We validated efficient FOXM1 and/or RHNO1 knockdown in OVCAR8 DR-GFP cells (*Figure 16D, E*). In contrast to U2OS in OVCAR8 cells both FOXM1 and RHNO1 knockdown impaired HR, but RHNO1 knockdown had a more substantial effect (*Figure 16F*). Similar to U2OS cells, dual FOXM1 + RHNO1 knockdown led to a greater reduction in HR capacity, but the effect

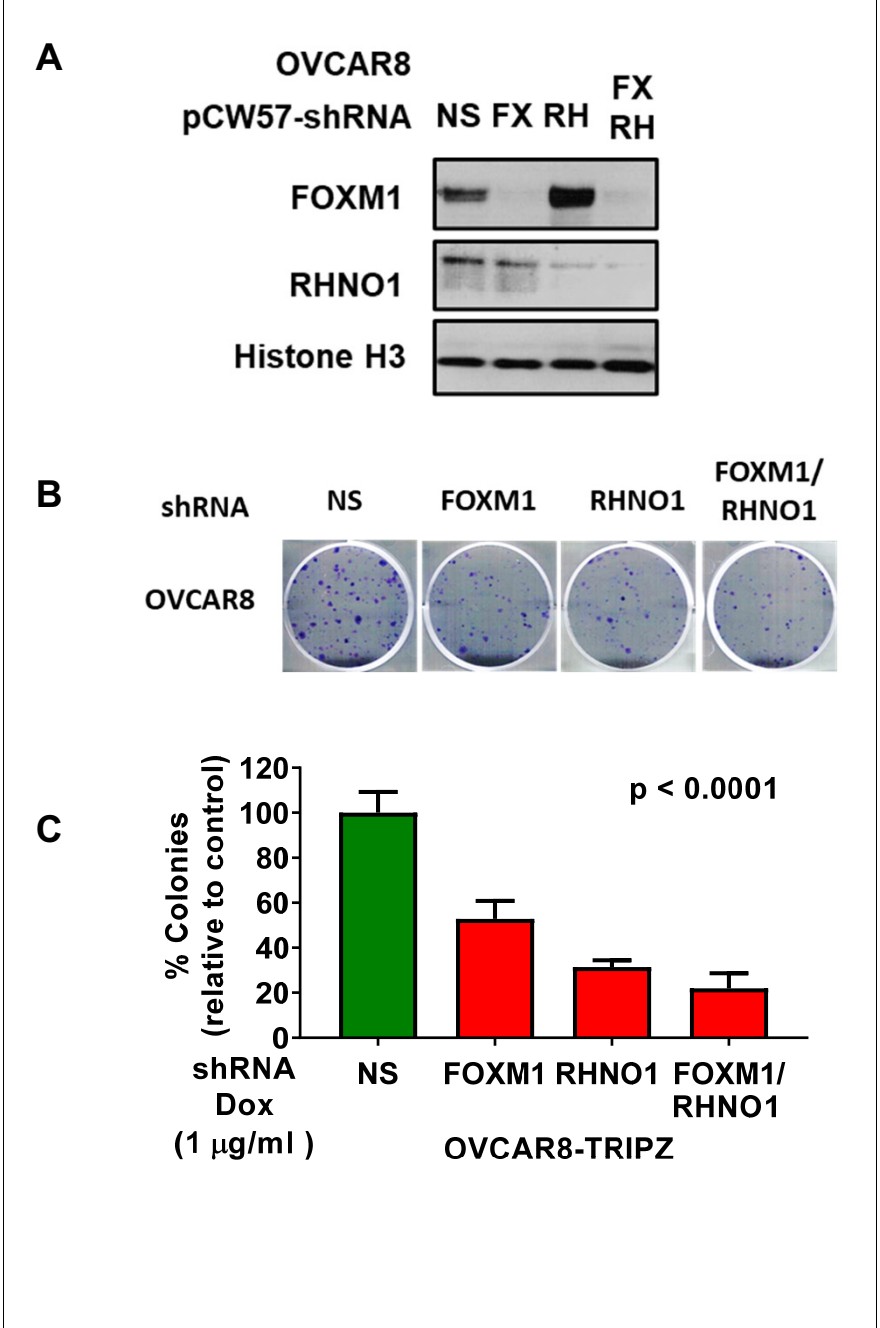

**Figure 12.** OVCAR8 cells with dual FOXM1 and RHNO1 shRNA knockdown have diminished clonogenic growth. OVCAR8 high-grade serous carcinoma cells were engineered for doxycycline (dox)-inducible FOXM1 and/or RHNO1 shRNA knockdown. (**A**) OVCAR8 cells treated with dox for 72 hr followed by western blot analysis. (**B**) Representative clonogenic growth data. Cells were seeded into a 6-well dish, in triplicate, at a density of 500 cells. Dox was added at the time of seeding, and media containing dox was replenished every 48 hr. Clonogenic survival was measured at 12 days, and cells were fixed with methanol and stained with crystal violet. (**C**) Quantification of clonogenic growth data. Colonies containing more than 50 cells were counted, and clonogenic survival was quantified as an average of the replicates. NS: nontargeting shRNA. The ANOVA linear trend p-value is shown.

The online version of this article includes the following figure supplement(s) for figure 12:

**Figure supplement 1.** CAOV3 cells with dual FOXM1 and RHNO1 shRNA knockdown have diminished clonogenic growth.

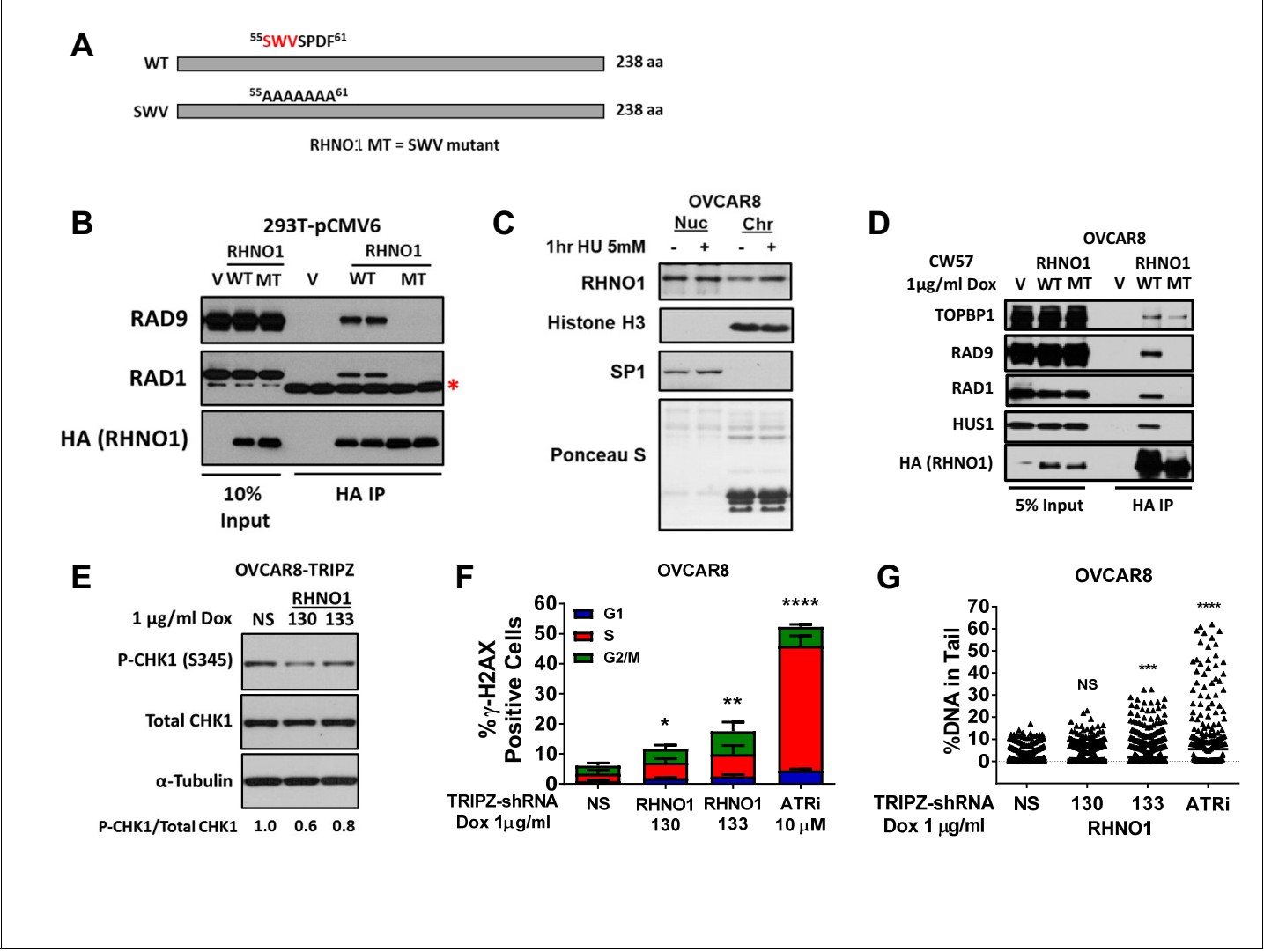

**Figure 13.** RHNO1 functions in the replication stress response in high-grade serous carcinoma cells. (**A**) Schematic of RHNO1 wild-type and SWV mutant. The proteins are the same molecular weight but residues 55–61 are converted to alanine in the SWV mutant, disrupting its ability to interact with 9-1-1 proteins. (**B**) 293T cells were transfected with empty vector, vector expressing HA-RHNO1-WT or SWV mutant. Protein was harvested 24 hr post transfection for Co-IP and western blot analyses. The red asterisk indicates a non-specific band. (**C**) OVCAR8 cells were seeded for 24 hr, then treated with vehicle or 5 mM hydroxyurea (HU) for 1 hr, and cells were harvested for subcellular fraction into soluble nuclear and chromatin bound proteins and western blot analysis. Histone H3 was used as the chromatin control and Sp1 as the soluble nuclear control. (**D**) OVCAR8-inducible RHNO1 knockdown cells were engineered for inducible expression of HA-tagged RHNO1-WT or SWV mutant, grown in the presence of doxycycline (dox) for 72 hr, and then protein was harvested for Co-IP/western blot analyses. (**E**) OVCAR8 western blot measurement of P-Chk1 sustaining RHNO1 shRNA knockdown. (**F**) OVCAR8 cells were treated with the ATR inhibitor VE-822 or dox to induce RHNO1 shRNA knockdown, and gH2AX-positive cells in different cell cycle phases were quantified by fluorescence-activated cell sorting. (**G**) COMET analysis of DNA strand breakage in OVCAR8 cells following the treatments described in (**F**).

was similar to that observed with RHNO1 knockdown alone (*Figure 16C, F*). Together, these data validate a significant role for RHNO1 in HR and show that FOXM1 also contributes to HR, potentially in cooperation with RHNO1, in a cell context-specific manner.

## FOXM1 and RHNO1 promote PARPi resistance in HGSC cells

Approximately half of HGSC tumors show HR deficiency, which confers PARPi sensitivity through a synthetic lethal interaction (*The Cancer Genome Atlas Research Network, 2011*; *Farmer et al., 2005*; *Lord et al., 2015*). HR-proficient HGSC are less sensitive to PARPi, which limits the clinical utility of these drugs (*Karakashev et al., 2020*). Based on DR-GFP data (*Figure 16*), we hypothesized

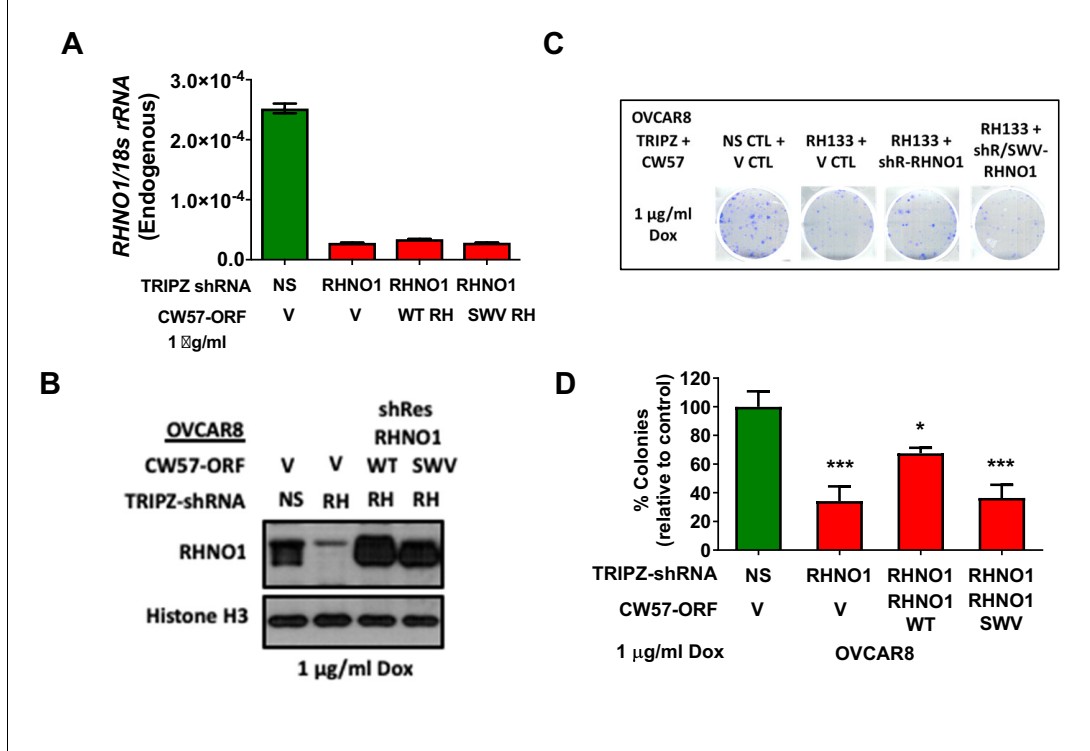

**Figure 14.** Binding to the 9-1-1 complex is required for RHNO1 function in OVCAR8 cell clonogenic growth. OVCAR8 cells were engineered for doxycycline (dox)-inducible RHNO1 knockdown and simultaneous dox-inducible expression of HA-RHNO1 WT or SWV mutant. (**A, B**) Cells were seeded in the presence of dox and grown for 72 hr to harvest RNA and protein extracts, followed by (**A**) RT-qPCR or (**B**) western blot to confirm knockdown efficiency. (**B**) also shows HA-RHNO1 protein ectopic expression efficiency. (**C**) Cells were seeded into 6-well dishes in triplicate, in the presence of dox and at a density of 500 cells. Media containing dox was replenished every 48 hr. Clonogenic survival was measured 12 days later. A representative data set is shown. (**D**) For the experiment described in (**C**), colonies containing more than 50 cells were counted and clonogenic survival was quantified as an average of the replicates. V: vector control; NS: nontargeting shRNA. t-test p-values are shown. p-value designation: ****<0.0001, ***<0.001, **<0.01, *<0.05.

that FOXM1 and RHNO1 may promote PARPi resistance in HGSC cells. To test this, we utilized OVCAR8 cells engineered for dox-inducible knockdown of FOXM1 and/or RHNO1. OVCAR8 cells, which have monoallelic *BRCA1* methylation but still express BRCA1 protein, display an intermediate HR phenotype and are functional for HR in DR-GFP assays (*Figure 16*; *Huntoon et al., 2013*). We utilized the prototype FDA-approved PARPi olaparib (*Kim et al., 2015*). Cell proliferation and cell survival assays revealed that either FOXM1 or RHNO1 knockdown sensitized OVCAR8 cells to olaparib, while dual knockdown led to a significantly greater increase in olaparib sensitivity (*Figure 17A, B*). In agreement, dual knockdown cells showed the highest level of DNA damage, assessed by γ-H2AX levels, as well as the greatest degree of G2/M arrest (*Figure 17C, D*).

As FOXM1 and RHNO1 are co-regulated by the F/R-BDP (*Figure 10*), we next determined whether F/R-BDP activity impacts olaparib sensitivity. For this task, we used a CRISPRi strategy in OVCAR8 cells (*Figure 10A*). Notably, CRISPRi directed to the F/R-BDP reduced the expression of both *FOXM1* and *RHNO1* and significantly sensitized OVCAR8 cells to olaparib, as measured by cell viability assays and clonogenic assays (*Figure 18A–C*). In addition, F/R-BDP repression increased olaparib-mediated apoptosis and G2/M arrest (*Figure 18D, E*). These data validate FOXM1 and RHNO1 as contributors to the response of HGSC cells to olaparib and, importantly, link this function to F/R-BDP activity.

## FOXM1 and RHNO1 promote acquired PARPi resistance

Although HR-deficient HGSC tumors are initially sensitive to PARPi, they often acquire PARPi resistance (*Lord et al., 2015*; *Bitler et al., 2017*; *Fojo and Bates, 2013*; *Sonnenblick et al., 2015*). To assess the possible role of FOXM1 and RHNO1 in acquired PARPi resistance, we used the

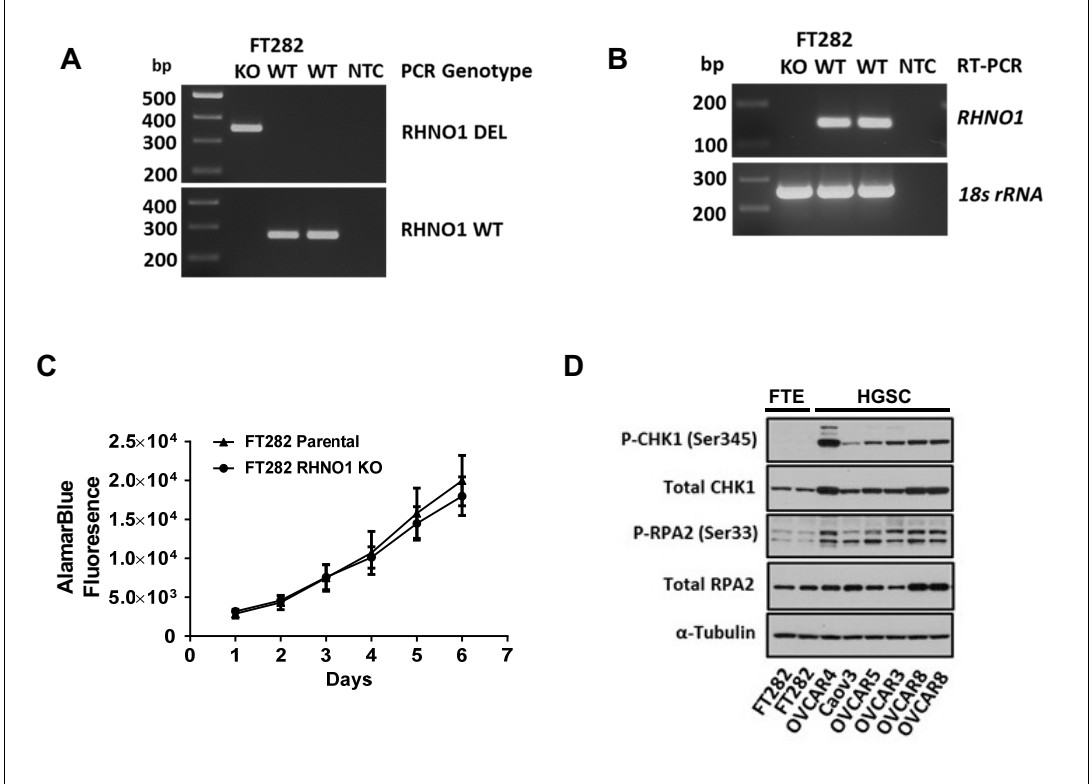

**Figure 15.** Homozygous knockout of RHNO1 in FT282 cells does not impact cell proliferation, and high-grade serous carcinoma (HGSC) cells show elevated expression of replication stress (RS) biomarkers compared to Fallopian tube epithelium (FTE) cells. RHNO1 knockout was achieved using CRISPR-Cas9 with guide RNAs that targeted near the start and stop codon of *RHNO1*. (**A, B**) *RHNO1* homozygous knockout was confirmed by (**A**) genomic DNA PCR and (**B**) *RHNO1* mRNA expression measured by end-point RT-PCR. NTC: no template control. (**C**) FT282 parental and RHNO1 clonal knockout cells were seeded in quadruplicate and grown in culture for a period of 1 week. Cell growth/viability was measured by the AlamarBlue assay every 24 hr. (**D**) Western blot analyses of two canonical phosphoprotein biomarkers of RS in FTE and HGSC cell lines. Cell line sample names are shown below.

UWB1.289 (UWB1) cell model system. UWB1 cells are BRCA1 null and highly sensitive to PARPi (*DelloRusso et al., 2007*; *Burgess et al., 2020*), while SyR12 and SyR13 are two UWB1 cell clones selected in vitro for acquired olaparib resistance (*Yazinski et al., 2017*). Notably, SyR clones are dependent on ATR for cell survival in the presence of PARPi (*Yazinski et al., 2017*). We engineered SyR12 and SyR13 cells to express CRISPRi directed to the F/R-BDP, as described above for OVCAR8 cells (*Figure 19A*). In agreement with OVCAR8 cell data, F/R-BDP repression sensitized both SyR12 and SyR13 cells to olaparib-mediated loss of cell viability (*Figure 19B, C*). While F/R-BDP repression did not restore olaparib sensitivity to the level observed in parental UWB1 cells (*Figure 19B, C*), the magnitude of sensitization is quite similar to that seen following treatment with ATRi (*Yazinski et al., 2017*).

## FOXM1 and RHNO1 promote carboplatin resistance

In addition to PARPi, HR-deficient tumors are sensitive to platinum drugs, a front-line chemotherapy for HGSC (*Bowtell et al., 2015*). In addition, FOXM1 has previously been shown to promote platinum resistance in ovarian cancer (*Zhang et al., 2014*; *Brückner et al., 2021*), and we observed several chemoresistant recurrent HGSC patients that showed coordinate upregulation of *FOXM1* and *RHNO1* (*Figure 8C*). We tested whether FOXM1 and RHNO1 influences the HGSC cellular response to carboplatin. Clonogenic assays of OVCAR8 cells revealed that dual FOXM1 + RHNO1 knockdown markedly enhanced the effects of carboplatin, while single-knockdown cells showed a substantial, but lesser, effect (*Figure 20*). These data further implicate FOXM1 and RHNO1 in the response of ovarian cancer cells to commonly utilized ovarian cancer chemotherapy.

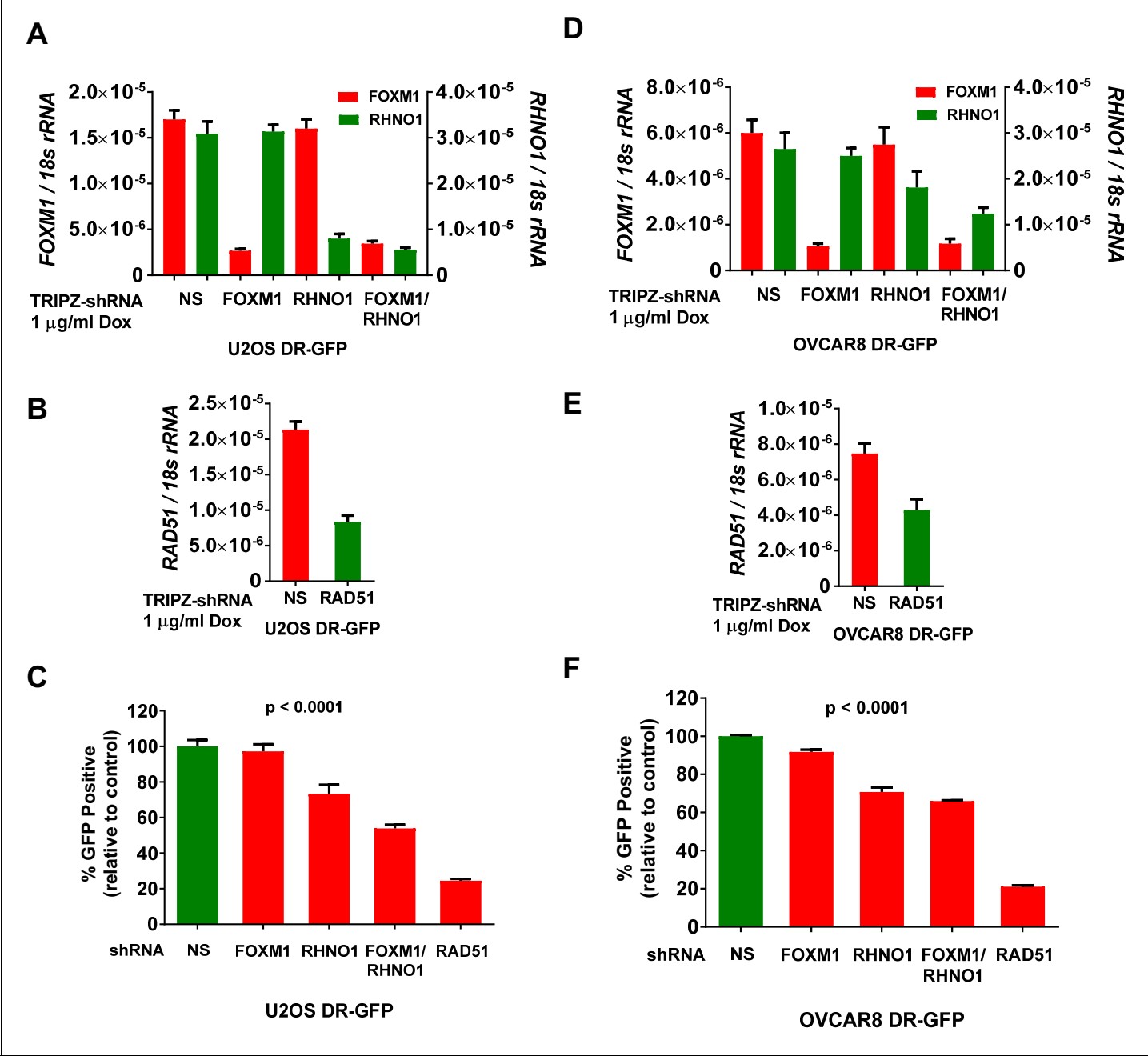

**Figure 16.** FOXM1 and RHNO1 promote homologous recombination (HR). DR-GFP cell lines were treated with doxycycline (dox) to induce shRNA expression, and mRNA knockdowns were measured by RT-qPCR. (A, B) *FOXM1*, *RHNO1*, and *RAD51* mRNA expression in U2OS-DR-GFP cells sustaining mRNA knockdown. (C) HR repair of I-SceI-induced DSBs in U2OS cells after the indicated mRNA knockdowns. Values are expressed relative to the percentage of green fluorescent protein (GFP)+ cells in I-SceI-transfected control cells. RAD51 knockdown was used as a robust positive control for HR impairment. (D, E) *FOXM1*, *RHNO1*, and *RAD51* mRNA expression in OVCAR8-DR-GFP cells. (F) HR repair of I-SceI-induced DSBs in OVCAR8 cells after the indicated mRNA knockdowns. Values are expressed relative to the percentage of GFP+ cells in I-SceI-transduced control cells. RAD51 knockdown was used as a robust positive control for HR impairment. All results are shown as mean ± SE from three independent replicates. (C) and (F) show p-values for the linear trend (excluding the RAD51 control).

## Discussion

We report that the well-known oncogene *FOXM1* is arranged in a head-to-head bidirectional configuration with *RHNO1*, a gene that promotes ATR-Chk1 signaling (*Cotta-Ramusino et al., 2011*;

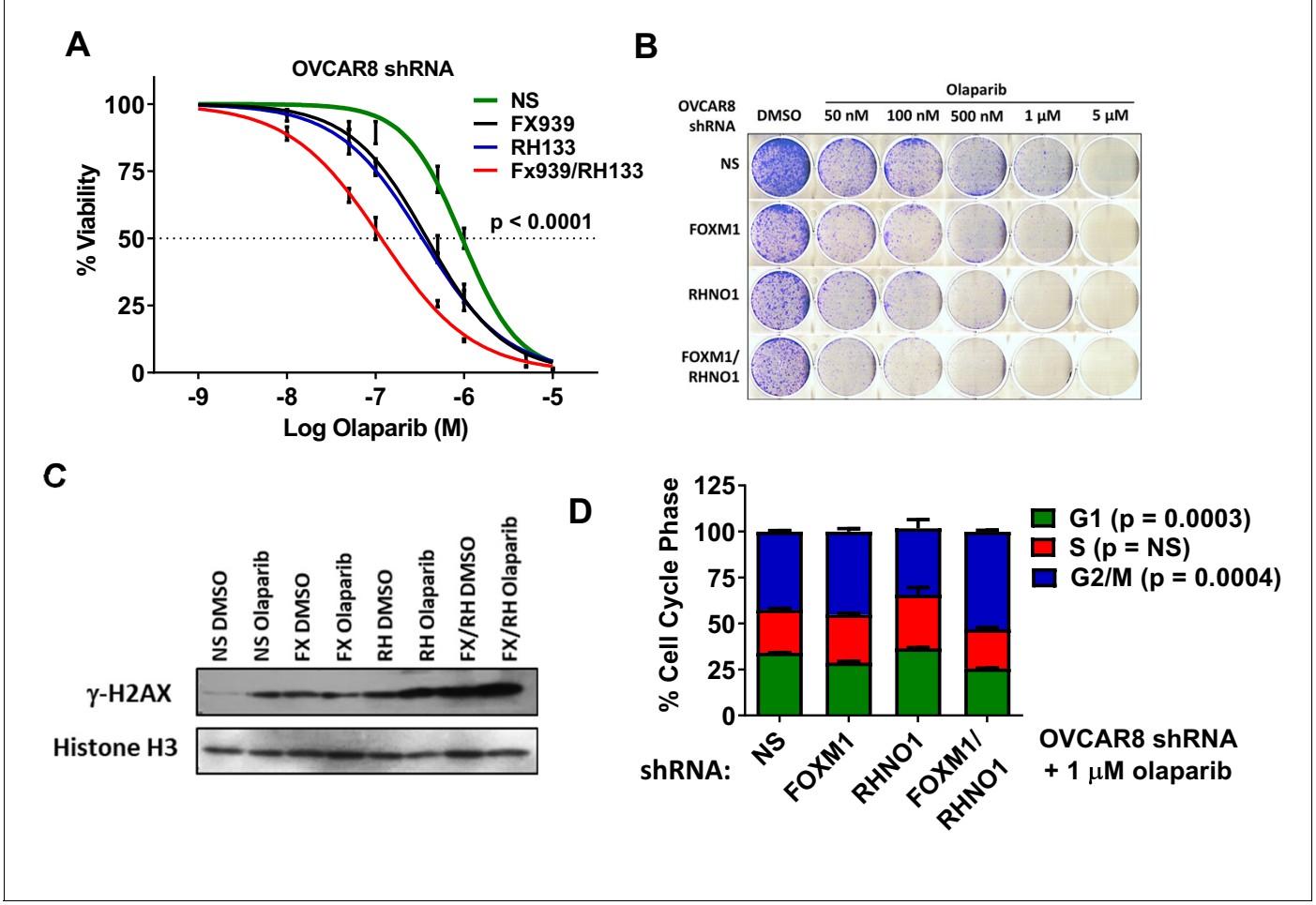

**Figure 17.** FOXM1 and RHNO1 knockdown-sensitize OVCAR8 cells to olaparib. OVCAR8 cells engineered for doxycycline (dox)-inducible FOXM1 and/ or RHNO1 shRNA knockdown were grown in dox for 72 hr prior to seeding cells for experiments. (A) Cells were seeded in 96-well plates in quadruplicate in the presence of dox at a density of 500 cells per well. 24 hr later, cells received media containing dox + vehicle control or olaparib, and this was repeated every 48 hr. Cell viability was measured at 8 days post-olaparib treatment using AlamarBlue assays. The ANOVA linear trend p-value is shown. (B) Cells were seeded into 6-well dishes in triplicate in the presence of dox at a density of 5000 cells per well. 24 hr later, cells received media containing dox and vehicle or olaparib, and this was repeated every 48 hr. After 8 days of growth, cells were fixed with methanol and stained with crystal violet for clonogenic assays. A representative result is shown. NS/CTL: nontargeting shRNA. (C) Western blot analysis of γ-H2AX in chromatin extracts from cells with dox-inducible control shRNA (NS), FOXM1 shRNA (FX), RHNO1 shRNA (RH), or both (FX/RH). After 48 hr in dox, cells were treated with vehicle control dimethyl sulfoxide (DMSO) or 1 μM olaparib for 24 hr, and protein extracts were harvested. Histone H3 served as a loading control. (D) Cells were treated as in (C), and cell cycle analyses were conducted 24 hr post-olaparib treatment.
The online version of this article includes the following figure supplement(s) for figure 17:

**Figure supplement 1.** Quantification of γ-H2AX in OVCAR8 chromatin extracts following FOXM1 and/or RHNO1 knockdown and DMSO or olaparib treatment.

*Lindsey-Boltz et al., 2015*). *FOXM1* and *RHNO1* mRNA expression correlate in both normal tissues and cancer, including HGSC, and is one of the most highly correlated BDG pairs genome-wide in FTE and HGSC cell models. These observations raise the possibility that some phenotypes characteristic of tumors with elevated FOXM1 expression might be mediated, in part, by RHNO1. Additionally, our data showing frequent genomic amplification of *RHNO1* in HGSC is consistent with recent observations that DDR genes are frequently amplified in human cancer (*Wu et al., 2020*). More broadly, our findings support that co-expressed BDG pairs functionally contribute to oncogenesis. Recently, it was reported that CNAs are less frequent at BDG as compared to single genes (*Thompson et al., 2018*). This suggests that the frequent copy number gains observed at the *FOXM1/RHNO1* BDG are under positive selection and may play an important role in oncogenesis.

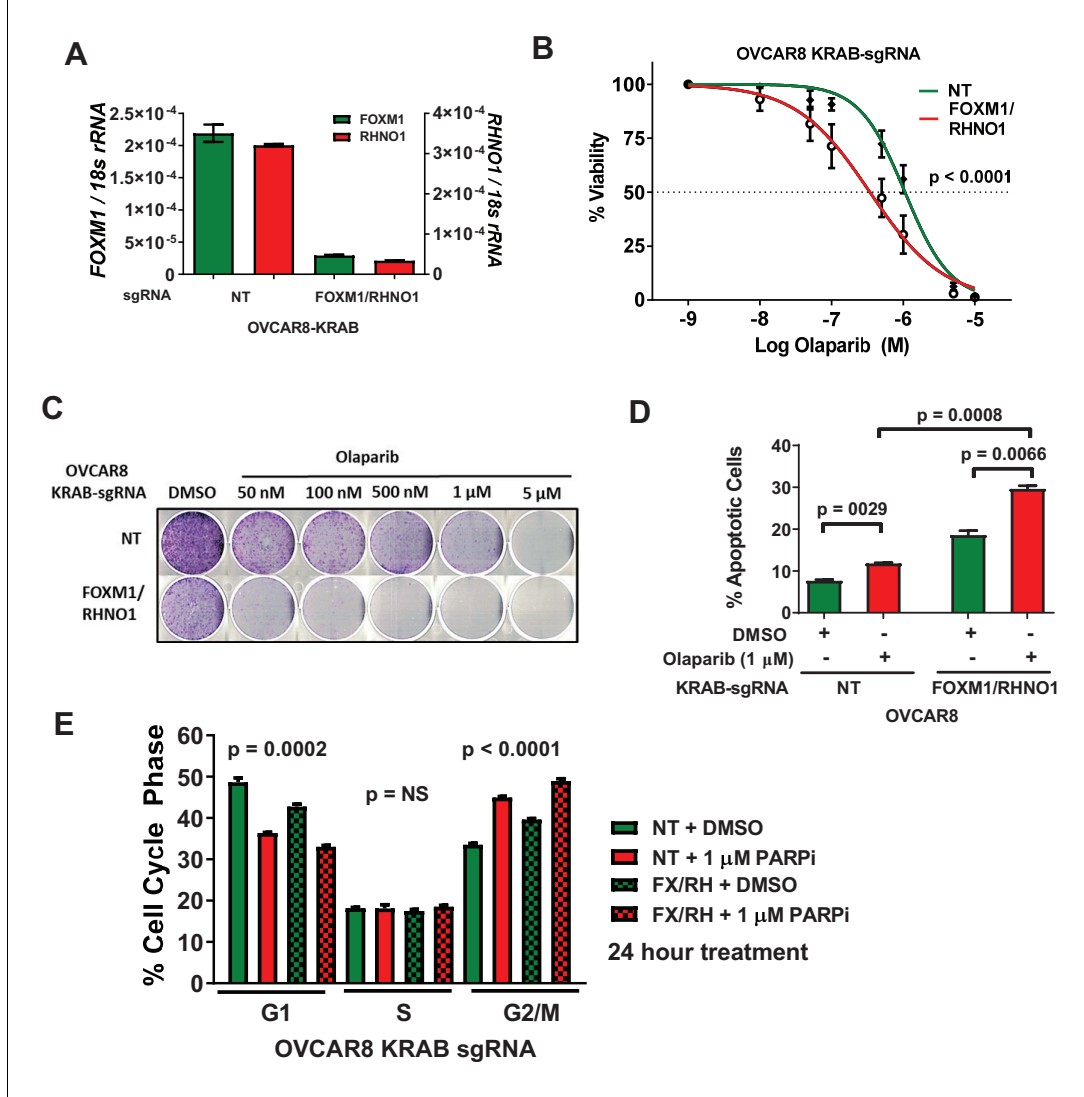

**Figure 18.** CRISPRi repression of the F/R-BDP sensitizes OVCAR8 cells to olaparib. (**A**) OVCAR8 cells expressing the KRAB transcriptional repressor and a guide RNA targeting the F/R-BDP, and the corresponding changes in mRNA expression as measured by RT-qPCR. NT: nontargeting guide RNA. (**B**) Cells were seeded in 96-well plates in quadruplicate at a density of 500 cells per well. 24 hr later, cells received media containing vehicle or olaparib, and this was repeated every 48 hr. Cell viability was measured at 8 days using AlamarBlue, and the IC50 for olaparib was determined. An F-test was used to compare IC50 values. (**C**) Cells were seeded into a 6-well dish, in triplicate, at a density of 5000 cells per well. 24 hr later, cells received media containing vehicle or olaparib, and this was repeated every 48 hr. After 8 days of growth, cells were fixed with methanol and stained with crystal violet to visualize colonies. (**D**) The indicated cells were treated with olaparib for 48 hr, and apoptosis was measured using Annexin V staining/fluorescence-activated cell sorting (FACS) analysis. (**E**) Cell cycle profiles were measured using propidium iodide staining/FACS analysis. ANOVA linear trend p-values are shown.

We experimentally validated that the intergenic space separating the *FOXM1* and *RHNO1* TSS functions as a BDP and contributes to *FOXM1* and *RHNO1* co-expression. While the transcription factors that regulate F/R-BDP activity are unknown, BDPs are enriched in specific transcription factor motifs, including GABPA, MYC, E2F1, E2F4, NRF-1, CCAAT, and YY1 (*Lin et al., 2007*). Of these, MYC and E2F1 are likely candidates for F/R-BDP regulation, as these factors are known to activate *FOXM1* expression (*Barger et al., 2019*; *Barger et al., 2015*; *Pan et al., 2018*; *Bollu et al., 2020*), and the F/R-BDP contains motifs for these TFs. Whether specific transcription factors symmetrically or asymmetrically regulate F/R-BDP activity is a key question as the *FOXM1/RHNO1* expression ratio increased in cancer vs. normal. Notably, scRNA-seq of both immortalized FTE cells and HGSC cells revealed that *FOXM1* and *RHNO1* are co-expressed in individual cells. Moreover, *FOXM1* and

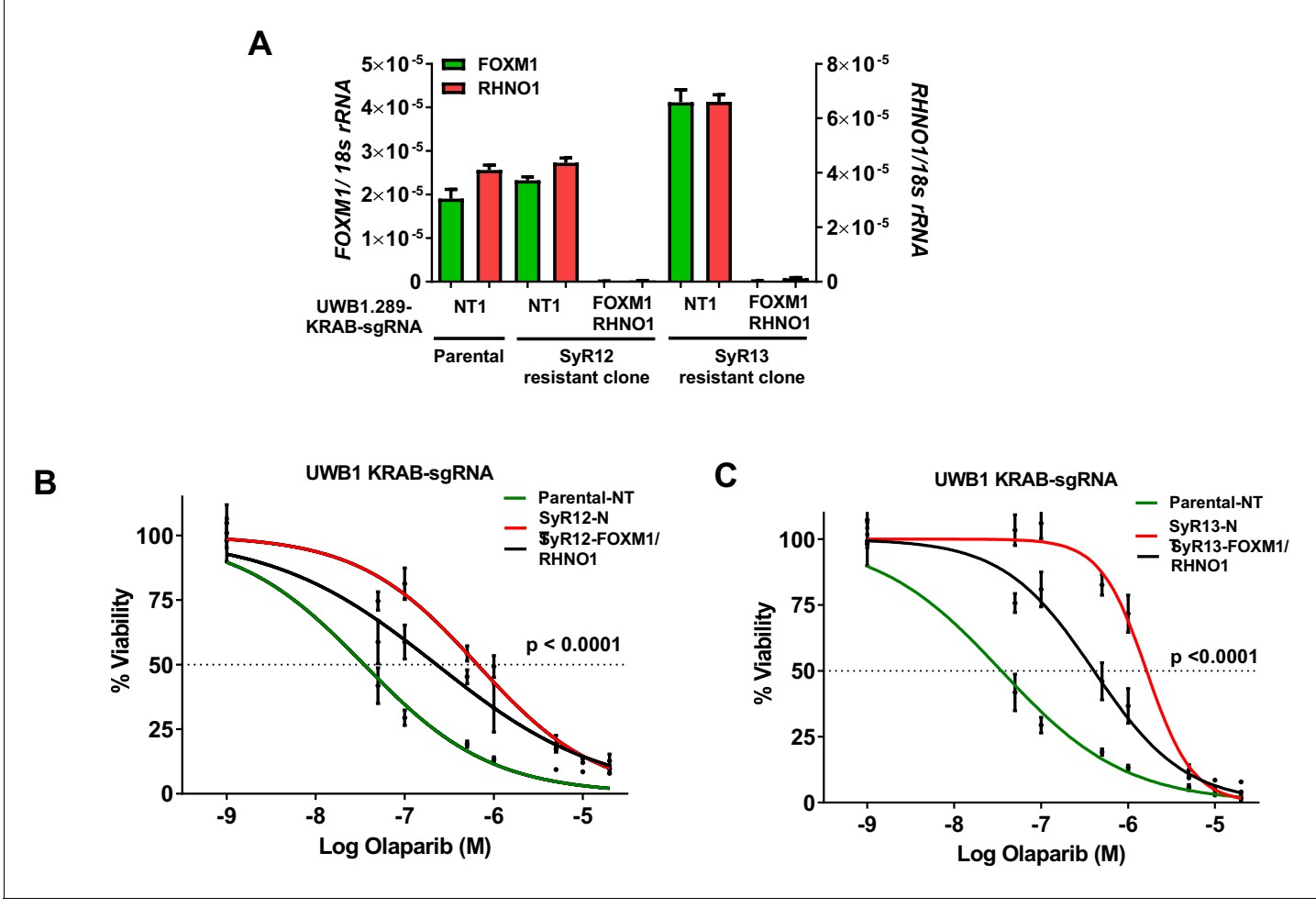

**Figure 19.** CRISPRi repression of the F/R-BDP in olaparib-resistant cells increases olaparib sensitivity. Parental and olaparib-resistant UWB1 cells were engineered to express the KRAB transcriptional repressor and a control guide RNA or guide RNA targeting the F/R-BDP (sg130). (A) Cells were seeded and grown for 72 hr, and RNA was harvested for RT-qPCR to measure CRISPRi efficiency. (B, C) UWB1 parental KRAB and (B) SyR12 KRAB or (C) SyR13 KRAB cells were seeded in 96-well plates in quadruplicate at a density of 750 cells per well. 24 hr later, cells received media containing vehicle or olaparib, and this was repeated every 48 hr. Cell viability was measured at 8 days post-treatment using AlamarBlue assays, and the IC50 for olaparib was determined. NT1: nontargeting control guide RNA. FOXM1/RHNO1: sg130 targeting the F/R-BDP. In (B, C), ANOVA linear trend statistics for the IC50 values are shown. Linear trends were ordered – SyR12/13 NT, SyR12/13 FOXM1/RHNO1 sg130, parental NT.

*RHNO1* was one of the highest correlated BDG pairs genome-wide. This, along with conservation of the F/R-BDP in vertebrates, suggests that FOXM1 and RHNO1 function cooperatively in normal physiology and cancer.

HGSC tissues and cells overexpress both *FOXM1* and *RHNO1*, suggesting oncogenic functions for each gene. While FOXM1 protein is overexpressed in solid tumors (*Halasi and Gartel, 2013a*; *Li et al., 2017b*; *Dai et al., 2015*), analogous studies of RHNO1 protein expression have not been reported. However, the present study demonstrates that individual targeting of either FOXM1 or RHNO1, using shRNA or CRISPR-Cas9, significantly reduces HGSC cell clonogenic growth, suggesting that each protein contributes to cancer phenotypes. In the case of RHNO1, our data show that it promotes HR, clonogenic growth, and chemotherapy resistance in HGSC cells. Furthermore, rescue experiments showed that the clonogenic growth phenotype promoted by RHNO1 is dependent on its binding to the 9-1-1 complex, in agreement with earlier characterizations of RHNO1 function (*Cotta-Ramusino et al., 2011*; *Lindsey-Boltz et al., 2015*; *Hara et al., 2020*). In contrast to HGSC cells, CRISPR-mediated knockout of RHNO1 in immortalized FTE cells did not result in growth defects. These data are consistent with the idea that cancer cells, which have elevated RS, have an increased dependence on RHNO1 for survival compared to normal cells. In agreement, we show

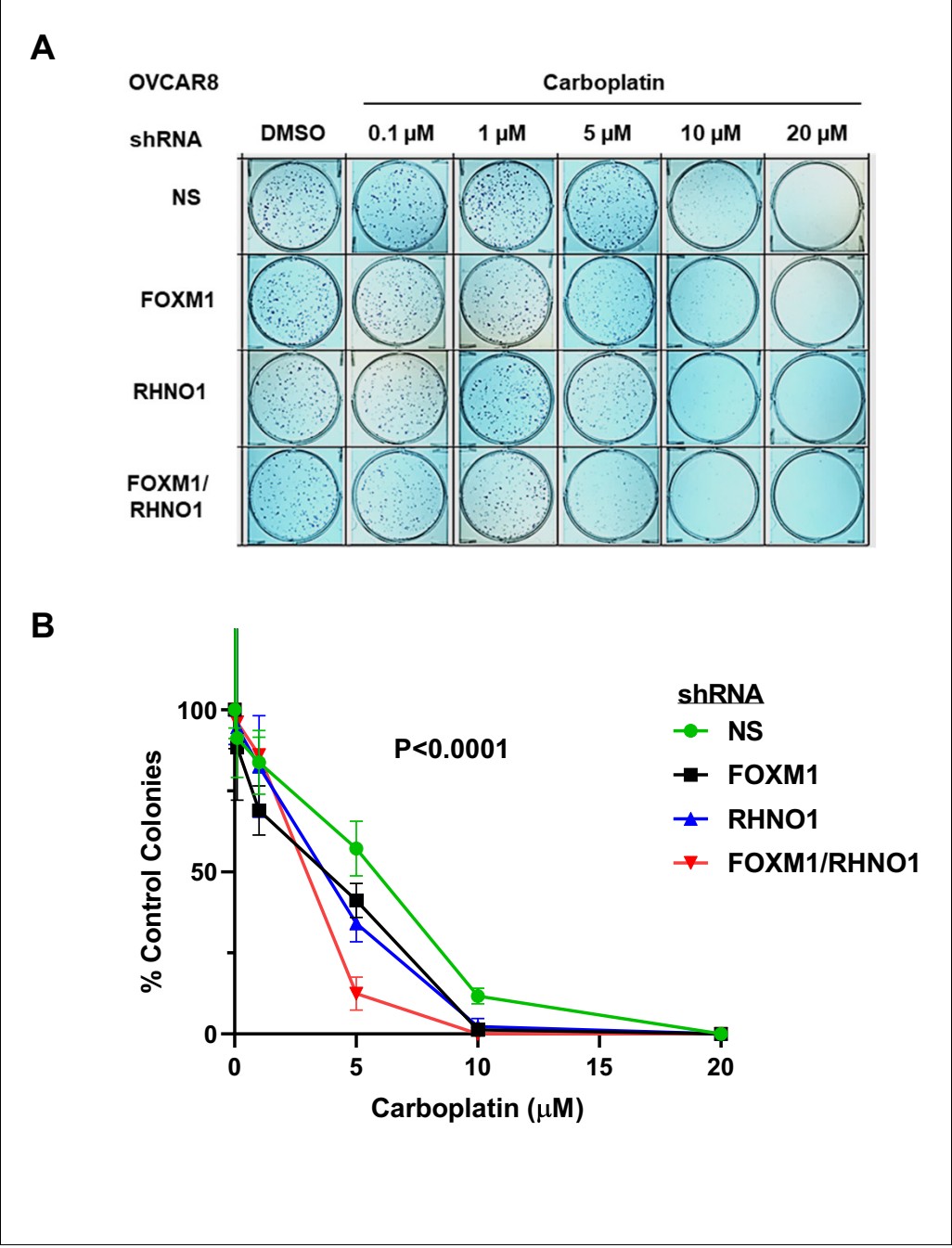

**Figure 20.** FOXM1 and RHNO1 knockdown enhances OVCAR8 cell sensitivity to carboplatin. OVCAR8 cells were treated with doxycycline (dox) for 48 hr and seeded into triplicate wells of 6-well dishes in single-cell suspension. Cells were treated with carboplatin and dox the day after seeding and every 48 hr thereafter, and allowed to form colonies for 7–21 days. After incubation, cells were fixed in methanol and stained using 0.5% crystal violet; colonies containing >~50 cells were counted. (**A**) Representative clonogenic growth data. (**B**) Quantified data for experimental groups. Each data set was normalized to the DMSO-only treatment data point. The ANOVA p-value for the group comparison is shown.

that HGSC cell lines have high elevation of canonical RS biomarker expression as compared to immortalized FTE cells.

While FOXM1 knockdown in HGSC cells reduced HR capacity, bulk RNA-seq data suggested that FOXM1 does not strongly regulate DNA repair genes in HGSC cell lines (data not shown).

Nonetheless, FOXM1 transcriptional regulation of DNA repair is complex and includes both direct and indirect gene targets (*Zona et al., 2014*). Thus, comprehensive gene expression analyses using several HGSC models will be required to help resolve the role of FOXM1 transcriptional activity in DNA repair. The DR-GFP assay may underestimate the impact of FOXM1 on HR as FOXM1 knockdown can lead to a partial G2/M arrest, which may indirectly increase HR capacity (*Hustedt and Durocher, 2016*). Future experiments using cell cycle synchronizations will be important to address the precise role of FOXM1 and RHNO1 in HGSC cell cycle progression and its relation to HR capacity.

Consistent with their functions in HR, we show that FOXM1 and RHNO1 jointly promote PARPi resistance both in treatment-naive HGSC cells and in HGSC cells with acquired PARPi resistance. Interestingly, dual FOXM1 + RHNO1 knockdown led to increased olaparib sensitivity as measured using cell viability, G2/M accumulation, and clonogenic growth assays compared to single-gene knockdown. One possible explanation for these data is that RHNO1 may inhibit FOXM1-mediated S/G2 progression, a recently described novel cell cycle checkpoint that is enforced by ATR (*Saldivar et al., 2018*). Thus, in the absence of RHNO1, due to reduced ATR/Chk1 activity, cells may progress more readily into G2/M, resulting in increased mitotic DNA damage (*Cotta-Ramusino et al., 2011*). Loss of RHNO1 leads to a less efficient RS response and increases DNA damage, while additional loss of FOXM1 might cause mitotic defects that increase the DNA-damaging effects of olaparib.

In addition to PARPi, FOXM1 and RHNO1 may impact the response to other chemotherapy. Both *FOXM1* and *RHNO1* were coordinately upregulated in approximately half of patients with chemoresistant recurrent HGSC, where the first-line therapy was carboplatin + paclitaxel. FOXM1 was reported to promote resistance to both of these drugs, and RHNO1, which promotes the DDR, might also augment carboplatin resistance (*Tassi et al., 2017*; *Kwok et al., 2010*; *Carr et al., 2010*; *Nestal de Moraes et al., 2015*; *Wang et al., 2013*). In agreement, our data support a role for both proteins in decreasing HGSC cell sensitivity to carboplatin. Although there are currently no inhibitors of RHNO1, ATR, Chk1, and/or WEE1 inhibitors might help impair the oncogenic function of RHNO1. In contrast to the clinically advanced RS response inhibitors, to date, FOXM1 inhibitors (FOXM1i) have only been used in preclinical studies (*Kalathil et al., 2020*; *Gormally et al., 2014*; *Ziegler et al., 2019*; *Brückner et al., 2021*; *Gartel, 2008*).

In some instances, we measured cell phenotypes downstream of dual FOXM1 + RHNO1 targeting and compared these to single knockdown. We note that the cellular response to dual FOXM1 + RHNO1 targeting is predicted to be complex due to the pleiotropic nature of FOXM1 function (*Kalathil et al., 2020*). For example, FOXM1 contributes to several cell cycle transitions including G1/S, S/G2, G2/M, and mitotic progression (*Halasi and Gartel, 2013a*; *Costa, 2005*; *Laoukili et al., 2005*; *Barger et al., 2015*; *Saldivar et al., 2018*; *Arceci et al., 2019*; *Zhang et al., 2016*; *Chen et al., 2013*; *Tan et al., 2010*; *Xue et al., 2010*). Thus, one model would predict that cells with reduced FOXM1 would have a reduced rate of inherited DNA damage caused by RHNO1 loss (*Cotta-Ramusino et al., 2011*) due to reduced cell cycle transit of cells with damaged DNA. This model seems to be in agreement with recent reports showing that FOXM1 sensitizes tumor cells to Chk1 and WEE1 inhibitors (*Chung et al., 2019*; *Branigan et al., 2021*; *Diab et al., 2020*). In contrast, because FOXM1 is also required for proper mitotic progression and disruption of FOXM1 increases mitotic DNA damage (*Laoukili et al., 2005*), it is plausible that targeting FOXM1 could, in some instances, augment the effects of RHNO1 depletion. The latter idea is consistent with the observation that decreased levels of FOXM1 cause mitotic decline, genomic instability, and senescence during human aging (*Macedo et al., 2018*). This model is more consistent with our data as dual targeting of FOXM1 and RHNO1 mostly caused additive reductions in HR activity and clonogenic growth capacity, and was not antagonistic. Finally, reduced cell cycle progression and cell proliferation seen in FOXM1 knockdown cells might lead to additive effects with RHNO1 knockdown, which reduces cell survival, particularly in clonogenic growth assays (*Barger et al., 2015*; *Kim et al., 2010*). In summary, the outcome of FOXM1/RHNO1 dual targeting, and how it compares to individual gene targeting, likely depends on the specific cellular context and cell phenotype examined.

A synopsis of our findings and its potential implications for HGSC biology and therapy are presented (*Figure 21*). *FOXM1* and *RHNO1* are BDG that are co-amplified in HGSC and other tumors due to focal amplification at 12p13.33 (*Barger et al., 2019*; *Barger et al., 2015*; *Paracchini et al., 2020*). *FOXM1* and *RHNO1* are co-expressed as a result of co-regulation by the F/R-BDP and are overexpressed in HGSC and numerous other cancers (*Barger et al., 2019*; this study). The

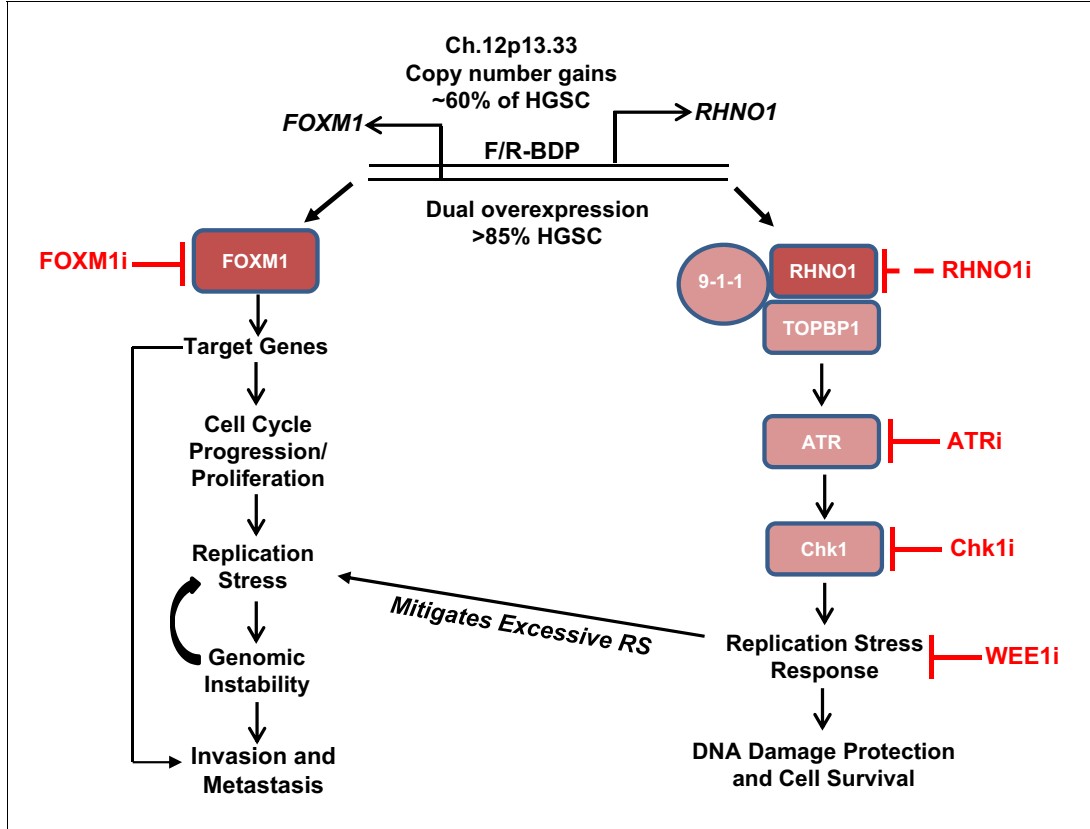

**Figure 21.** Model for FOXM1/RHNO1 bidirectional gene (BDG) function and cooperativity. *FOXM1* and *RHNO1* are BDGs regulated by a bidirectional promoter (BDP) and are co-amplified and co-expressed in high-grade serous carcinoma (HGSC), including single cells. Both genes are overexpressed in HGSC and other cancers. FOXM1, via the activation of specific gene targets, promotes oncogenic phenotypes, including cell cycle progression, proliferation, replication stress (RS), genomic instability, and metastatic progression; the latter phenotype may reflect activation of a distinct set of transcriptional targets later in tumor progression. In contrast, RHNO1 promotes ATR-Chk1 signaling via its binding to 9-1-1 and TOPBP1 and promotes the RS response. This protects HGSC cells from excessive RS and DNA damage and enhances cell survival. RHNO1 co-expression may help to mitigate excessive FOXM1-induced RS and thus facilitate other FOXM1 oncogenic functions. FOXM1 and RHNO1 co-expression may increase tumor resiliency by allowing tumor cells to gain a fitness advantage of increased growth and proliferation without accumulating excessive DNA damage. Consistently, FOXM1-expressing tumors are highly enriched for biomarkers of RS (*Li et al., 2020*). Therapeutic targeting of the FOXM1/RHNO1 bidirectional gene unit may be accomplished using inhibitors of FOXM1 and/or ATR/Chk1/WEE1. RHNO1 inhibitors have not been reported (dashed line).

oncogenic functions of FOXM1 in HGSC include cell cycle progression, genomic instability, tumor invasion, and metastatic spread (*Parashar et al., 2020*; *Barger et al., 2019*; *Barger et al., 2015*). On the other hand, RHNO1 may promote oncogenesis by increasing the efficiency of the ATR-Chk1-dependent RS response, which protects cells from excessive DNA damage and promotes cell survival (*Kim et al., 2010*; *Cotta-Ramusino et al., 2011*; *Lindsey-Boltz et al., 2015*).

What is the functional link between FOXM1 and RHNO1? We observed that ectopic expression of FOXM1 in FTE cells increases RS biomarker expression and alters replication fork dynamics (unpublished data). Thus, FOXM1 may induce RS, while RHNO1 responds to RS. Critical support for this model is provided by a recent study that demonstrated that FOXM1 expression induced RS and genomic instability in bronchial epithelial cells and U2OS cells (*Li et al., 2020*). Additionally, the authors showed that FOXM1 expression correlates with RS biomarkers in The Cancer Protein Atlas data from several cancer types, including HGSC (*Li et al., 2020*). We thus hypothesize that the mechanistic link between FOXM1 and RHNO1 centers around RS, in that RHNO1 may help to mitigate excessive RS driven by FOXM1 (*Figure 21*). FOXM1 and RHNO1 co-expression may provide a selective advantage to tumors by enabling a balance between increased proliferation and cell cycle progression, which promotes RS, and the RS response, which limits DNA damage and promotes cell survival. Co-activation of these two proteins may thus increase tumor resiliency, that is, allowing

tumor cells to grow and survive in the face of challenges including hypoxia, DNA-damaging chemotherapy, immune cell infiltration, and other intrinsic and extrinsic insults.

Finally, regarding the timing of these events in ovarian cancer, it is known that FOXM1 is expressed in HGSC precursor lesions in the FTE, which may, along with other proteins such as cyclin E1, increase RS, thus providing a selective advantage for RHNO1 co-expression (*Karst et al., 2014*; *Levanon et al., 2014*). Later during tumor progression, genomic instability functions as an important driver of continued RS independent of particular oncoproteins (*Passerini et al., 2016*; *Wilhelm et al., 2020*). However, FOXM1 remains highly expressed because it promotes critical functions in late-stage progression unrelated to RS and genomic instability (*Halasi and Gartel, 2013a*; *Kalathil et al., 2020*). Additionally, in late-stage tumors, RHNO1 and other RS response proteins such as ATR, Chk1, and WEE1 become critically important to restrict excessive DNA damage, which would otherwise impair cell survival. Overall, this conceptual framework may prove useful to guide future studies of FOXM1 and RHNO1 and their interplay in normal physiology and cancer.

## Materials and methods

### TCGA, CCLE, GTEx, and RIKEN FANTOM5 data retrieval

TCGA, CCLE, and GTEx data sets were retrieved from cBioPortal and University of California Santa Cruz (UCSC) Xena Browser. The genomic profiles of *FOXM1* and *RHNO1* were analyzed in the TCGA HGSC data set (Ovarian Serous Cystadenocarcinoma-TCGA Provisional), including somatic CNAs from GISTIC (*Beroukhim et al., 2007*) using Onco Query Language and mRNA expression using RNA-seq (RNA-seq V2 RSEM). *FOXM1* and *RHNO1* expression profiles from TOIL GTEx (cell lines removed) and TCGA HGSC RNA-seq data sets were obtained from UCSC Xena. *FOXM1* and *RHNO1* mRNA expression data from Cap Analysis of Gene Expression analysis of mouse tissues was obtained from the RIKEN FANTOM5 project (RNA-seq) data set obtained from http://www.ebi.ac.uk.

### UCSC Genome Browser data

The genomic region of *FOXM1* and *RHNO1* was retrieved from UCSC Genome Browser (http://genome.ucsc.edu) (*Kent et al., 2002*) using the human genome build hg19 and the genomic coordinates cHR12:2966150–2998815. Several tracks were selected and displayed, including *FOXM1* and *RHNO1* mRNA, CGI, DNAseI hypersensitivity, H3K4Me1, H3K4Me3, and H3K27Ac ChIP-seq tracks, transcription factor ChIP-seq, genome segmentations, and conserved genome tracks from 100 vertebrates and mammalian genomes. The Genome Browser screenshot tool was used to obtain images.

### Illumina Infinium HumanMethylation450K data

TCGA pan-cancer DNA methylation (HumanMethylation450K) data were downloaded from UCSC Xena (http://xena.ucsc.edu) consisting of 9753 samples. This data set contains the DNA methylation 450K array beta values compiled by combining available data from all TCGA cohorts. The GDSC1000 collection (The Genomics of Drug Sensitivity in Cancer Project, Welcome Sanger Institute) was downloaded from NCBI GEO (GSE68379) (*Iorio et al., 2016*). For all data sets, the DNA methylation profile was measured using the Illumina Infinium HumanMethylation450 platform and the Infinium HumanMethylation450 BeadChip Kit. The method consists of approximately 450,000 probes querying the methylation status of CpG sites within and outside CGI. The DNA methylation score of each CpG is represented as a beta value, which is a normalized value between 0 (fully unmethylated) and 1 (fully methylated).

### Sodium bisulfite DNA sequencing (bisulfite sequencing)

Bisulfite sequencing was used to determine the methylation status of the *F/R-BDP* region in NO and EOC human tissue samples (*Clark et al., 1994*). Genomic DNA was isolated using the Puregene Tissue Kit (QIAGEN) and was chemically converted using the EZ DNA Methylation Kit (Zymo Research Corp.). The F/R-BDP region containing a CGI was amplified from the bisulfite-converted DNA using methylation PCR-specific primers, designed using MethPrimer (*Supplementary file 2*; *Li and Dahiya, 2002*). Gradient PCR reactions were performed using a C1000 Touch Thermal Cycler (Bio-Rad) to optimize annealing temperatures. After gel purification using the QIAquick Gel Extraction

Kit (QIAGEN), PCR products were cloned using the TOPO TA Cloning Kit (Invitrogen). Between 9 and 15 individual clones were sequenced for each sample, and Sanger sequencing was performed by the UNMC DNA Sequencing Core Facility. DNA sequence information was analyzed using the Lasergene SeqMan Pro program (DNASTAR).

## Cell lines

COV362 and COV318 cell lines (Sigma) were cultured in DMEM (Corning) supplemented with 10% fetal bovine serum (FBS, Invitrogen), 2 mM glutamine (Life Technologies), and 1% penicillin-strepto-mycin (pen-strep, Life Technologies). KURAMOCHI and OVSAHO (Japanese Collection of Research Bioresources Cell Bank) and SNU-119 (Korean Cell Line Bank) cell lines were cultured in RPMI-1640 (Hyclone) supplemented with 10% FBS and 1% pen-strep. OVCAR4 cells (National Cancer Institute Division of Cancer Treatment and Diagnosis Cell Line Repository) were cultured in RPMI-1640 supplemented with 10% FBS and 1% pen-strep. OVCAR8 cells (National Cancer Institute Division of Cancer Treatment and Diagnosis Cell Line Repository) were cultured in DMEM (Corning) supplemented with 10% FBS and 1% pen-strep. OVCAR3 cells (American Type Culture Collection) were cultured in DMEM with 10% FBS and 1% pen-strep. CAOV3 and OVCAR5 cell lines were a gift from Anirban Mitra (Indiana University) and were cultured in DMEM with 10% FBS and 1% pen-strep. UWB1.289, UWB1-SyR12, and UWB1-SyR13 cells were cultured in RPMI 1640/Mammary Epithelial Growth Media (1:1; Hyclone/PromoCell) supplemented with 3% FBS, MEGM growth factors, and 1% pen-step (*Yazinski et al., 2017*). Primary hOSE cells (ScienCell) were cultured in Ovarian Epithelial Cell Medium (ScienCell, 7311). Human immortalized FTE cells (FT190, FT282-E1, and FT282) were cultured in DMEM-Ham's F12 50/50 (Corning) supplemented with 2% USG (Pall Corporation) or 10% FBS and 1% pen-strep (*Karst et al., 2014*; *Karst et al., 2011*). IOSE-T cells (IOSE-21, hOSE immor-talized with hTERT) (*Li et al., 2007*) were a gift from Francis Balkwill (Cancer Research UK) and were cultured in Medium 199/MCDB105 (1:1, Sigma) supplemented with 15% FBS, 1% pen-strep, 10 ng/mL human epidermal growth factor (Life Technologies), 0.5 µg/mL hydrocortisone (Sigma), 5 µg/mL bovine insulin (Cell Applications), and 34 µg protein/mL bovine pituitary extract (Life Technologies). IOSE-SV cells (IOSE-121, hOSE immortalized with SV40 Large T antigen) were a gift from Nelly Auer-sperg (University of British Columbia) and were cultured in Medium 199/MCDB105 (1:1) supplemented with 10% FBS and 25 µg/mL gentamicin (Life Technologies). HEK293T cells (American Type Culture Collection) were cultured in DMEM with 10% FBS and 1% pen-strep. U2OS-DR-GFP (282C) cells were a gift from Jeremy Stark (City of Hope) and were grown in McCoy's 5A (Corning), 10% FBS, and 1% pen-strep (*Gunn et al., 2011*; *Gunn and Stark, 2012*). OVCAR8-DR-GFP cells were a gift from Larry Karnitz and Scott Kaufmann (Mayo Clinic, Rochester) and were grown in the same conditions as OVCAR8 parental cells (*Huntoon et al., 2013*). All cell lines were maintained at 37°C in a humidified incubator with 5% $CO_2$. Cell culture medium was changed every 3–5 days depending on cell density. For routine cell passage, cells were split at a ratio of 1:3 to 1:10 when they reached 85to–90% confluence. Doxycycline-inducible cells were treated every 48 hr with doxycycline unless otherwise noted.

Stocks of all cell lines were authenticated using short tandem repeat analysis at the DNA Services Facility, University of Illinois at Chicago. All cell line stocks were confirmed to be mycoplasma free using PCR analyses at the Epigenomics Core Facility, University of Nebraska Medical Center. Cells thawed for experimental use were discarded prior to 1 month post thaw.

## Reverse transcription quantitative PCR (RT-qPCR)

Total RNA was purified using TRIzol (Invitrogen) and RNA was DNase-treated using the DNA-free kit (Ambion) or Direct-zol RNA Purification Kit (Zymo Research) with in-column DNase treatment. RNA quality was determined by running samples on RNA denaturing gels (1.2% agarose gel in 1X MOPS buffer containing formaldehyde) and checking for RNA degradation. DNase-treated RNA was con-verted to cDNA using the iScript cDNA Synthesis Kit (Bio-Rad). 1 µL of 1:5 cDNA sample dilutions were used for qPCR reactions. Standard curves were prepared using gel-purified end-point RT-PCR products. All samples were run in triplicate using the CFX Connect Real-Time System (Bio-Rad), and all gene expression data were normalized to 18S rRNA. PCR was performed using an annealing tem-perature of 60°C and a total of 45 cycles for all primer pairs. Dissociation curves were performed to confirm specific product amplification. Gradient PCRs were performed using the C1000 Touch

Thermal Cycler (Bio-Rad) to determine the annealing temperatures for each primer set. Primer sequences are listed in *Supplementary file 2*. Primer sequences were designed using NCBI Primer Blast (*Ye et al., 2012*) or were selected from those previously reported in the literature.

## Western blotting

Whole-cell protein extracts were prepared using radio-immunoprecipitation assay (RIPA) buffer (1× PBS, 1% NP40, 0.5% sodium deoxycholate, 0.1% sodium dodecyl sulfate [SDS]) supplemented with protease and phosphatase inhibitors (Sigma). Briefly, cells were lysed for 10 min in RIPA buffer and then sonicated, and the resulting extracts were centrifuged at 4°C for 10 min at 14,000 *g* to remove cell debris. Nuclear protein extracts were prepared using the NE-PER Nuclear and Cytoplasmic Extraction Kit (Thermo Scientific) supplemented with protease and phosphatase inhibitors. Protein concentrations were determined using the BCA protein assay (Thermo Scientific). Equal amounts of protein (~30–50 μg) were fractionated on 4–12% gradient SDS-polyacrylamide gel electrophoresis gels (Invitrogen) and transferred to PVDF membrane (Roche). Membranes were stained with Ponceau S to confirm efficient transfer and equal loading, and were then blocked with 5% nonfat dry milk in Tris-buffered saline Tween-20 (TBST) for 1 hr at room temperature. The membranes were then incubated with primary antibodies in 5% nonfat dry milk or 5% bovine serum albumin (BSA) in TBST at 4°C overnight followed by incubation with secondary antibodies in 5% nonfat dry milk in TBST for 1 hr at room temperature. Antibody information is provided in the Key resources table. Enhanced chemiluminescence (Thermo Scientific) was used for protein detection. Quantification of protein expression was performed using ImageJ software (Image Processing and Analysis in Java, National Institute of Health) (*Schneider et al., 2012*).

## Tissues

Human NO tissues and EOC tissues were obtained from patients undergoing surgical resection at Roswell Park Comprehensive Cancer Center (RPCCC) using Institutional Review Board-approved protocols as described previously (*Akers et al., 2014*). Pathology specimens were reviewed at RPCCC, and tumors were classified according to World Health Organization criteria (*Serov et al., 1973*). Genomic DNA and total RNA extractions of tissues were performed as described previously (*Woloszynska-Read et al., 2008*).

## Single-cell RNA sequencing

FT282 and OVCAR8 cells were seeded on day 0 at ~50% and ~30% confluency, respectively, and harvested on day 2 at ~80% confluence. Cells were trypsinized and cell suspensions were pipetted several times and passed through a 40 μm cell strainer to ensure a single-cell suspension. Cell viability (Trypan blue exclusion) was confirmed to be >90% using the TC20 Automated Cell Counter (Bio-Rad). Cell pellets were washed twice in PBS containing 0.04% BSA, resuspended in PBS, and 0.04% BSA to ~500,000 cells/mL, and transported to the UNMC DNA Sequencing Core Facility for single-cell capture, library preparation, and scRNA-seq. As per manufacturer's instructions, ~2750 cells were loaded per channel to achieve a target of ~1500 captured cells. Single cells were loaded on a CHRomium Single Cell Instrument (10X Genomics) to generate single-cell gel bead in emulsions (GEMs) for the partitioning of samples and reagents into droplets. GEMs contain oligos, lysed cell components, and Master Mix. GEMs were processed as follows: cells were lysed, RNA extracted, and reverse transcribed to generate full-length, barcoded cDNA from the poly A-tailed mRNA transcripts. Barcoded cDNA molecules from each cell were PCR-amplified in bulk followed by enzymatic fragmentation. The Qubit was then used to measure and optimize the insert size of the double-stranded cDNA prior to library construction. Single-cell cDNA and RNA-Seq libraries were prepared using the Chromium Single Cell 3′ Library and Gel Bead Kit v2 (10x Genomics), and the products were quantified on the 2100 Bioanalyzer DNA High Sensitivity Chip (Agilent). Each fragment contains the 10x Barcode Unique Molecular Identifier (UMI) and cDNA insert sequence used for data analysis. During library construction, Read 2 is added by Adapter ligation. Both single-cell libraries were sequenced using the Illumina NextSeq 550 system using the following parameters: pair-end sequencing with single indexing, 26 cycles for Read 1 and 98 cycles for Read 2. FT282 cells produced a total of 85.6 million reads and OVCAR8 produced 82 million reads, and both had an overall Q30 for the run of 87.6%. Sequencing data were transferred to the Bioinformatics and Systems

Biology Core Facility at UNMC for analysis. The Cell Ranger Single Cell Software (10X Genomics) was used to process raw bcl files to perform sample demultiplexing, barcode processing, and single-cell 3′ gene counting (https://software.10xgenomics.com/single-cell/overview/welcome). Each data set was aligned to a combined human (hg19) reference, and only cells that were identified as aligned to human in the 'filtered' output of the Cell Ranger count module were used for analysis.

The statistical method scImpute was used to accurately and robustly impute the transcript drop-outs (zero values) that exist in scRNA-seq data from sequencing small amounts of RNA. The scImpute R package was downloaded from https://github.com/Vivianstats/scImpute; *Li and Li, 2018* and was run in R with default settings. FT282 and OVCAR8 scRNA-seq raw gene expression matrix files (.mtx) were converted into dense expression matrices using 'cellRanger mat2csv' command. The scImpute input data consisted of a scRNA-seq matrices with rows representing genes and columns representing cells, and output data consisted of an imputed count matrix with the same dimension. FT282 and OVCAR8 scRNA-seq data were deposited into the GEO database (GSE150864).

BDG pairs were obtained by searching the UCSC database for protein-coding gene pairs in which the TSS diverge on opposite DNA strands and are located within 1000 base pairs. For each cell line, we computed the Pearson's correlation coefficient between all BDG pairs. We performed principal component analysis (PCA) and computed correlation coefficients on the data sets using Python.

## scRNA-seq data set retrieval

scRNA-seq data sets from mouse normal colon epithelium (GSE92332) (*Haber et al., 2017*) and human melanoma (GSE72056) (*Tirosh et al., 2016*) were downloaded from NCBI GEO. scRNA-seq data for human HGSC tissues were obtained from a published data set (*Winterhoff et al., 2017*). scRNA-seq data were normalized using scImpute (*Li and Li, 2018*).

## *FOXM1* and *RHNO1* expression in primary and recurrent HGSC

We obtained *FOXM1* and *RHNO1* RNA-seq data from patient-matched primary and recurrent HGSC from the European Genome-phenome Archive (EGA) https://ega-archive.org/, using EGAD00001000877 and EGAD00010001403 (*Patch et al., 2015*; *Kreuzinger et al., 2017*).

## Promoter activity luciferase reporter assays

Luciferase reporter assays were performed using the pGL4 Luc Rluc Empty (Firefly and Renilla luciferase) and pCMV6-SEAP (SEAP, transfection control) plasmid constructs. Firefly and Renilla luciferase activities were measured 24 hr after transfection using the Dual-Luciferase Reporter Assay System (Promega) and a GloMax 20/20 luminometer (Promega). Luciferase activity was expressed in arbitrary units as displayed on the luminometer after a 10 s integration time. SEAP activity was measured 24 hr after transfection using the Phospha-Light kit (Thermo Scientific) and a POLARstar OPTIMA microplate reader (BMG Labtech). Transfections were performed in triplicate in each individual experiment.

## Promoter activity GFP/RFP reporter assays (FACS)

Single-cell F/R-BDP bidirectional promoter activity assays were performed using the pTurbo-RFP-GFP plasmid containing a cloned F/R-BDP promoter region. The pTurbo-RFP-GFP empty plasmid was used as the negative control. pTurbo plasmids containing the PGK promoter and either GFP or RFP alone were used as fluorescence-activated cell sorting (FACS) compensation controls. Briefly, 900,000 HEK293T cells were seeded into 6-well dishes and cells were transfected 24 hr later using Lipofectamine 2000 (Life Technologies) and 2 µg of plasmid DNA in Opti-MEM. Cells were harvested 24 hr after transfection, washed with PBS, and 10,000 events per sample were analyzed by FACS to determine GFP and RFP expression.

## RLM-RACE mapping

The TSS of *FOXM1* and *RHNO1* were determined using the FirstChoice RLM-RACE Kit (Ambion) according to the manufacturer's instructions. RNA was isolated using TRIzol reagent (Invitrogen) followed by purification with Direct-zol RNA MiniPrep with in-column genomic DNA digestion. RLM-RACE allows for the specific amplification of 5′ capped RNA, which is found only in full-length mRNA. The specific outer and inner (nested) primers in combination with adaptor-specific primers

were used for amplifying the 5′ ends of *FOXM1* and *RHNO1* mRNA (*Supplementary file 2*). As negative controls, non-tobacco alkaline phosphatase (TAP)-treated aliquots from each RNA source were used; in all cases, this did not yield specific product amplification (data not shown). In contrast, *FOXM1*- and *RHNO1*-specific PCR products of various sizes were yielded from the TAP-containing reactions. PCR products were separated on 2% agarose gels, excised, and purified using the QIAquick gel extraction kit (QIAGEN). Gel-purified PCR products were cloned using the TOPO TA Cloning Kit (Invitrogen), and individual clones were Sanger sequenced by the UNMC DNA Sequencing Core Facility. DNA sequence information was analyzed using the Lasergene SeqMan Pro program (DNASTAR).

## Production of lentiviral particles and lentiviral transductions

Replication-deficient lentivirus was produced by transient transfection of 6.0 µg psPAX2 (Addgene #12260), 2.0 µg pMD2.G (Addgene #12259), and 8.0 µg transfer plasmid into HEK293T cells in a 10 cm dish with Lipofectamine 2000 reagent (Life Technologies) according to the manufacturer's instructions. Viral supernatants were collected 48 hr post-infection and passed through a 0.2 µm filter. Functional titration was performed by transduction of cells with serially diluted virus in the presence of polybrene (4 µg/mL, Sigma) for 6 hr, followed by puromycin, G418, blasticidin, or hygromycin (Life Technologies) selection for 48 hr post-infection. For dual knockdown and rescue experiments requiring two different lentiviral constructs, cell lines underwent sequential transduction and selection.

## CRISPR-Cas9 gene knockouts

CRISPR-Cas9-mediated knockout was performed as previously described (*Ran et al., 2013*). Briefly, Lipofectamine 2000 (Life Technologies) was used to transfect HEK293T cells with PX458 (Addgene #48138), which expresses Cas9 and a guide RNA. *Supplementary file 2* contains a list of all sgRNA used. Guide RNAs were designed using the Genetic Perturbation Platform Web Portal (https://portals.broadinstitute.org/gpp/public/), and the highest scoring guide RNAs were empirically tested for their cutting efficiency. Isolation of clonal cell lines was achieved with FACS followed by plating of GFP-positive cells into 96-well dishes for clonal growth. DNA was isolated from clones using Quick-Extract (Epicentre) for genotyping and replica plated for clonal expansion. Genotyping was performed by PCR with an annealing temperature of 60°C and 30 cycles for all primer pairs listed in *Supplementary file 2*. Primers were designed with NCBI Primer-BLAST, then empirically tested by gradient end-point PCR to optimize specificity and sensitivity.

 *FOXM1* and *RHNO1* lentiCRISPR gene editing efficiency was assessed by PCR using Phusion High-Fidelity DNA polymerase (Thermo Fisher Scientific) followed by Sanger sequencing. TIDE analysis was performed to decompose the Sanger sequencing data and quantify the insertions and deletions (indels) in the DNA of the targeted cell pools (*Brinkman et al., 2014*). TIDE analysis was performed using the default settings, and parental genomic DNA was used as the control.

## RNA interference and plasmid constructs

 *Supplementary file 2* lists the miR-30-based short hairpin RNAs used (shRNAs, Dharmacon). The Key resources table lists all plasmids used. shRNAs were expressed from the TRIPZ dox-inducible lentiviral plasmid (Dharmacon), which co-expresses RFP and puromycin. The miR-30 shRNA cassettes containing RFP-shRNA were subcloned from TRIPZ into pCW57 (pCW57-MCS1-P2A-MCS2-Hygro, Addgene# 80922; pCW57-MCS1-P2A-MCS2-Blast, Addgene# 80921; and pCW57-MCS1-P2A-MCS2-Neo, Addgene# 89180) using the restrictions enzymes AgeI and MluI to generate an RNAi system with additional selection markers beyond puromycin (hygromycin, neomycin, and blasticidin). All plasmids were sequence verified. Plasmids were transfected with Lipofectamine 2000 (Life Technologies) according to the manufacturer's instructions.

## Drug treatments

Detailed information on the drugs used in this study (i.e., olaparib, berzosertib, hydroxyurea, etoposide, carboplatin, and doxycycline) is given in the Key resources table. Specific drug concentrations and time points used in experiments are described in the figures and legends.

## Clonogenic growth (i.e., colony formation) assays

Cells were trypsinized, counted, and seeded at a density of 500–1500 cells per well in triplicate wells of 6-well dishes in single-cell suspension and allowed to form colonies for 7–21 days. Following incubation, cells were fixed in methanol and stained using 0.5% crystal violet in PBS, rinsed with water, and air-dried overnight. Colonies containing >~50 cells were counted using inverted light microscope manually or from captured images using countPHICS (count and Plot HIstograms of Colony Size) software, a macro written for ImageJ (*Brzozowska et al., 2019*).

## Apoptosis assays

Cellular apoptosis was measured using the Muse Annexin V and Dead Cell kit (Millipore) according to the manufacturer's instructions. Briefly, the spent cell culture media containing floating cells was collected and saved, and adherent cells were harvested with trypsin-EDTA. Media containing 10% FBS was added to the trypsinized cells and then combined with the spent media and total cell numbers were counted. The cell suspension was centrifuged at $300 \times g$ for 5 min, and the cell pellet was resuspended to a final concentration of $1 \times 10^5$ cells/mL using media containing 1% FBS. Untreated cells were used as the negative control. Annexin V staining was achieved by first warming the Muse Annexin V and Dead Cell Reagent to room temperature before adding 100 µL to each tube of cell suspensions. The suspensions were gently and thoroughly mixed by pipetting up and down 10 times with a 1000 µL pipette. Samples were incubated for 20 min at room temperature in the dark. Immediately after incubation, the percentage of apoptotic cells was analyzed by flow cytometry using Muse Cell Analyzer (Millipore) system.

## Co-immunoprecipitation assays

Cells expressing empty vector and HA-tagged ORFs were lysed with M-PER (Pierce) containing Halt Protease and Phosphatase Cocktail (Pierce) and Nuclease S1. Lysates were mixed end-over-end at 4°C for 30 min, then centrifuged at 4°C for 10 min at 14,000 g to remove cell debris. Protein concentration were determined using the BCA protein assay (Thermo Scientific). Immunoprecipitations were performed using anti-HA magnetic beads and 500 µg total protein per sample. Samples were incubated overnight at 4°C with end-over-end mixing. The next day the magnetic beads were washed, and proteins were eluted using sample loading buffer. The entire sample was then loaded onto western blots. 5–10% total protein was used for input comparisons.

## FACS analyses of cell cycle and γ-H2AX expression

Cells were fixed in 70% ethanol overnight for cell cycle and/or γ-H2AX expression analysis of single cells. Fixed cells were washed with PBS and incubated overnight in PBS containing 1% BSA, 10% goat serum, and pS139-H2AX antibody (Millipore), then washed and incubated in goat anti-mouse Alexa Fluor 647 antibody for 30 min at room temperature. Cells were then incubated in 50 µg/mL propidium iodide and 100 µg/mL RNase A for 30 min, and 10,000 cells per sample were analyzed using a BD FACSarray (BD Biosciences) using 532 and 635 nm excitations and collecting fluorescent emissions with filters at 585/42 nm and 661/16 nm (yellow and red parameters, respectively). In other experiments, the γ-H2AX staining procedure was omitted and cells were only processed for cell cycle analyses using propidium iodide staining.

## Comet assays

The Comet Assay Kit (Trevigen) was used according to the manufacturer's instructions. Briefly, cells were suspended in low-melt agarose, layered onto treated slides to promote attachment (500 cells per slide), lysed, and subjected to electrophoresis (1 V/cm) under alkaline conditions to reveal single- and double-stranded DNA breaks using the Comet Assay Electrophoresis System II (Trevigen). Samples were then fixed, dried, and stained with SYBR Gold. Images were acquired with a ×10× objective lens using the EVOS FL Cell Imaging System (Thermo Fisher). Comet tail size was quantified using Comet Analysis Software (Trevigen).

## Homologous recombination repair (HR) assay (DR-GFP assay)

U2OS and OVCAR8 cells were previously generated for stable integration of DR-GFP, an HR substrate that generates a functional GFP upon successful HR after I-SceI cutting (*Gunn and Stark,*

*2012*; *Huntoon et al., 2013*; *Pierce et al., 1999*). U2OS and OVCAR8 cells were grown in the presence of doxycycline for 48 hr to induce shRNA expression and gene knockdown. U2OS cells were transfected with pCBASceI (Addgene; #26477) in the presence of 1 µg/mL doxycycline. Media containing doxycycline was changed 3 hr post-transfection, and cells were kept in culture for an additional 48 hr. OVCAR8 cells were transduced for 8 hr with AdNGUS24i, an adenovirus expressing I-SceI (a gift from Drs. Frank Graham and Phillip Ng, Baylor College of Medicine). Cells were harvested and fixed with 10% formaldehyde, then washed and analyzed by FACS to determine the fraction of GFP-positive cells.

## AlamarBlue cell viability assays

Cells were seeded at a density of 500–1000 cells per well into quadruplicate wells of sterile 96-well plates and treated as described in the figures and legends. AlamarBlue (Bio-Rad) was used to assess cell proliferation and cell viability. Background fluorescence values were obtained from wells containing only media and were subtracted from all data points. Data from drug-treated cells were normalized to vehicle-treated controls.

## Statistical analyses

The Student's t-test was used to compare differences between means of two groups. The Mann–Whitney test was used to compare differences between medians of two groups. The F-test was used to compare statistical models that were fitted to a data set by nonlinear regression. One-way analysis of variance (ANOVA) with a post-test for linear trend was used to compare two or more groups. For all analyses, significance was inferred at $p < 0.05$ and the computed p values were two-sided. GraphPad Prism (GraphPad Software, Inc) was used to perform statistical tests.

## Data deposit

Newly generated scRNA-seq data was deposited into the GEO database under accession number GSE150864.

# Acknowledgements

We thank Keith Johnson for original project insight and Gargi Ghosal for critical reading of the manuscript. We thank Anirban Mitra for CAOV3 and OVCAR5 cells, Larry Karnitz and Scott Kaufmann for OVCAR8 DR-GFP cells, Jeremy Stark and Tadayoshi Bessho for U2OS DR-GFP cells, Francis Balkwill for IOSE-T cells, Nelly Auersperg for IOSE-SV cells, Frank Graham and Philip Ng for SceI adenovirus, Stephen Elledge for the RHNO1-SWV mutant, and Aziz Sancar for RHNO1 antibody. We thank the UNMC Epigenomics, Genomics, Bioinformatics, and Flow Cytometry Core Facilities for expert assistance. This work was funded by the Rivkin Center for Ovarian Cancer, the Fred and Pamela Buffett Cancer Center, NCI P30 CA036727, NIH T32CA009476, NCI F99CA212470, NCI P50CA228991, the University of Nebraska Core Facility Grant Program, a UNMC Program of Excellence Assistantship, and the Betty J and Charles D McKinsey Ovarian Cancer Research Fund.

# Additional information

## Funding

| Funder | Grant reference number | Author |
| --- | --- | --- |
| National Institutes of Health | P30CA036727 | Adam R Karpf |
| Rivkin Center for Ovarian Cancer | 2018 Kirwin-Hinton Family Bridge Funding Award | Adam R Karpf |
| University of Nebraska Medical Center | | Carter J Barger |
| University of Nebraska Medical Center | | Adam R Karpf |
| National Institutes of Health | T32CA009476 | Carter J Barger |
| National Institutes of Health | F99CA212470 | Carter J Barger |

| National Institutes of Health | P50CA228991 | Ronny Drapkin |
|---|---|---|

The funders had no role in study design, data collection and interpretation, or the decision to submit the work for publication.

## Author contributions

Carter J Barger, Conceptualization, Data curation, Formal analysis, Funding acquisition, Investigation, Methodology, Writing - original draft, Writing - review and editing; Linda Chee, Catalina Munoz-Trujillo, Lidia Boghean, Connor Branick, Investigation; Mustafa Albahrani, Data curation, Investigation; Kunle Odunsi, Resources; Ronny Drapkin, Lee Zou, Resources, Writing - review and editing; Adam R Karpf, Conceptualization, Resources, Formal analysis, Supervision, Funding acquisition, Investigation, Writing - original draft, Project administration, Writing - review and editing

## Author ORCIDs

Adam R Karpf https://orcid.org/0000-0002-0866-0666

## Decision letter and Author response

Decision letter https://doi.org/10.7554/eLife.55070.sa1
Author response https://doi.org/10.7554/eLife.55070.sa2

# Additional files

## Supplementary files

- Supplementary file 1. *FOXM1* vs. *RHNO1* mRNA expression correlations.
- Supplementary file 2. Oligonucleotide sequences.
- Supplementary file 3. List of bidirectional gene pairs to support *Figure 5* (Excel file).
- Transparent reporting form

## Data availability

All data generated are found within the manuscript and supporting files. sc-RNA-seq data is deposited in GEO under accession number GSE150864.

The following dataset was generated:

| Author(s) | Year | Dataset title | Dataset URL | Database and Identifier |
|---|---|---|---|---|
| Barger CJ, Chee L, Albahrani M, Munoz-Trujillo C, Boghean L, Branick C, Odunsi K, Drapkin R, Zou L, Karpf AR | 2020 | Single cell RNA sequencing of human high-grade serous ovarian cancer (HGSC) cells and human immortalized fallopian tube epitheilal (FTE) cells | https://www.ncbi.nlm.nih.gov/geo/query/acc.cgi?acc=GSE150864 | NCBI Gene Expression Omnibus, GSE150864 |

The following previously published datasets were used:

| Author(s) | Year | Dataset title | Dataset URL | Database and Identifier |
|---|---|---|---|---|
| Abugessaisa I, Ramilowski JA, Lizio M, Severin J, Hasegawa A, Harshbarger J, Kondo A, Noguchi S, Yip CW, Ooi JL, Tagami M, Hori T, Agrawal S, Hon CC, Cardon M, Ikeda S, Ono H, Bono H, Kato M, Hashimoto | 2021 | RNAseq | https://fantom.gsc.riken.jp/5/datafiles/latest/ | RIKEN FANTOM5, FANTOM5 |

| | | | | | |
|---|---|---|---|---|---|
| K, Bonetti A, Kobayashi N, Shin J, Hoon M, Hayashizaki Y, Carninci P, Kawaji H, Kasukawa T | | | | | |
| Barretina J, Caponigro G, Stransky N, Venkatesan K, Margolin AA, Kim S, Wilson CJ, Lehár J, Kryukov GV, Sonkin D, Reddy A, Liu M, Murray L, Berger MF, Monahan JE, Morais P, Meltzer J, Korejwa A, Jané-Valbuena J, Mapa FA, Thibault J, Bric-Furlong E, Raman P, Shipway A, Engels IH, Cheng J, Yu GK, Yu J, Aspesi P, de Silva M, Jagtap K, Jones MD, Wang L, Hatton C, Palescandolo E, Gupta S, Mahan S, Sougnez C, Onofrio RC, Liefeld T, MacConaill L, Winckler W, Reich M, Li N, Mesirov JP, Gabriel SP, Getz G, Ardlie K, Chan V, Myer VE, Weber BL, Porter J, Warmuth M, Finan P, Harris JL, Meyerson M, Golub TR, Morrissey MP, Sellers WR, Schlegel R, Garraway LA | 2012 | Cancer Cell Line Encyclopedia | https://data.broadinstitute.org/ccle/CCLE_DepMap_18Q2_RNA-seq_RPKM_20180502.gct | CCLE, CCLE_DepMap_18Q2_RNAseq_RPKM_20180502.gct | |
| GTEx Consortium | 2017 | GTEx | https://toil-xena-hub.s3.us-east-1.amazonaws.com/download/gtex_RSEM_Hugo_norm_count.gz | GTEx, gtex_RSEM_Hugo_norm_count.gz | |
| TCGA Consortium | 2018 | TCGA Pan Cancer | https://tcga-pancan-atlas-hub.s3.us-east-1.amazonaws.com/download/EB%2B%2BAdjustPANCAN_IlluminaHiSeq_RNASeqV2.geneExp.xena.gz | TCGA Pan Cancer, EB%2B%2BAdjustPANCAN_IlluminaHiSeq_RNASeqV2.geneExp.xena.gz | |
| TCGA Consortium | 2011 | TCGA Ovarian Cancer | https://tcga-xena-hub.s3.us-east-1.amazonaws.com/download/TCGA.OV.sampleMap%2FHiSeqV2.gz | TCGA HGSC, TCGA.OV.sampleMap%2FHiSeqV2.gz | |
| Haber AL, Biton M, Rogel N, Herbst RH, Shekhar K, Smillie C, Burgin G, Delorey TM, Howitt MR, Katz Y, Tirosh I, Beyaz S, Dionne | 2017 | scRNA-seq intestine | https://www.ncbi.nlm.nih.gov/geo/query/acc.cgi?acc=GSE92332 | NCBI Gene Expression Omnibus, GSE92332 | |

| | | | | | |
|---|---|---|---|---|---|
| D, Zhang M, Raychowdhury R, Garrett WS, Rozenblatt-Rosen O, Ning Shi H, Yilmaz O, Xavier RJ, Regev A | | | | | |
| Tirosh I, Izar B, Prakadan SM, Wadsworth MH, Treacy D, Trombetta JJ, Rotem A, Rodman C, Lian C, Murphy G, Fallahi-Sichani M, Dutton-Regester K, Lin J, Cohen O, Shah P, Lu D, Genshaft AS, Hughes TK, Ziegler CGK, Kazer SW, Gaillard A, Kolb KE, Villani A, Johannessen CM, Andreev AY, Van Allen EM, Bertagnolli M, Sorger PK, Sullivan RJ, Flaherty KT, Frederick DT, Jané-Valbuena J, Yoon CH, Rozenblatt-Rosen O, Shalek AK, Regev A, Garraway LA | 2016 | scRNA-seq melanoma | | https://www.ncbi.nlm.nih.gov/geo/query/acc.cgi?acc=GSE72056 | NCBI Gene Expression Omnibus, GSE72056 |
| Patch AM, Christie EL, Etemadmoghadam D, Garsed DW, George J, Fereday S, Nones K, Cowin P, Alsop K, Bailey PJ, Kassahn KS, Newell F, Quinn MCJ, Kazakoff S, Quek K, Wilhelm-Benartzi C, Curry E', Leong HS, Australian Ovarian Cancer Study Group; Anne Hamilton, Mileshkin L, Au-Yeung G, Kennedy C, Hung J, Chiew YE, Harnett P, Friedlander M, Quinn M, Pyman J, Cordner S, O'Brien P, Leditschke J, Young G, Strachan K, Waring P, Azar W, Mitchell C, Traficante N, Hendley J, Thorne H, Shackleton M, Miller DK, Arnau GM, Tothill RW, Holloway TP, Semple T, Harliwong I, Nourse C, Nourbakhsh E, Manning S, Idrisoglu S, Bruxner | 2015 | HGSC RNA-seq | | https://ega-archive.org/datasets/EGAD00001000877 | European Genome-Phenome Archive, EGAD00001000877 |

| | | | | |
|---|---|---|---|---|
| TJC, Christ AN, Poudel B, Holmes O, Anderson M, Leonard C, Lonie A, Hall N, Wood S, Taylor DF, Xu Q, Fink JL, Waddell N, Drapkin R, Stronach E, Gabra H, Brown R, Jewell A, Nagaraj SH, Markham E, Wilson PJ, Ellul J, McNally O, Doyle MA, Vedururu R, Stewart C, Lengyel E, Pearson JV, deFazio A, Grimmond SM, Bowtell DDL | | | | |
| Kreuzinger C, Geroldinger A, Smeets D, Braicu EI, Sehouli J, Koller J, Wolf A, Darb-Esfahani S, Joehrens K, Vergote I, Vanderstichele A, Boeckx B, Lambrechts D, Gabra H, Wisman GBA, Trillsch F, Heinze G, Horvat R, Polterauer S, Berns E, Theillet C, Castillo-Tong DC | 2017 | HGSC RNA-seq | https://ega-archive.org/datasets/EGAD00010001403 | European Genome-Phenome Archive, EGAD00010001403 |

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

# Appendix 1

**Appendix 1—key resources table**

| Reagent type (species) or resource | Designation | Source or reference | Identifiers | Additional information |
|---|---|---|---|---|
| Gene (*Homo sapiens*) | FOXM1 | GenBank | Gene ID: 2305 | |
| Gene (*Homo sapiens*) | RHNO1 | GenBank | Gene ID: 83695 | |
| Antibody | FOXM1 (rabbit monoclonal) | Cell Signaling Technology | 5436 | (1:1000) |
| Antibody | β-Actin (mouse monoclonal) | Santa Cruz | sc-47778 | (1:200) |
| Antibody | α-Tubulin (rabbit polyclonal) | Cell Signaling Technology | 2144 | (1:1000) |
| Antibody | Lamin B (goat polyclonal) | Santa Cruz | sc-6217 | Discontinued |
| Antibody | Histone H3 (goat polyclonal) | Santa Cruz | sc-8654 | Discontinued |
| Antibody | CHEK1 (mouse monoclonal) | Santa Cruz | sc-8408 | (1:200) |
| Antibody | P-CHEK1 S345 (rabbit monoclonal) | Cell Signaling Technology | 2348 | (1:1000) |
| Antibody | P-CHEK1 S317 (rabbit monoclonal) | Cell Signaling Technology | 12302 | (1:1000) |
| Antibody | RPA32/RPA2 (rabbit monoclonal) | Abcam | ab76420 | (1:1000) |
| Antibody | RPA32 (S33) (rabbit polyclonal) | Bethyl | A300-246A-M | (1:1000) |
| Antibody | H2AX (goat polyclonal) | Bethyl | A303-837A | (1:2000) |
| Antibody | P-H2AX S139 (rabbit monoclonal) | Cell Signaling Technology | 9718 | (1:1000) |
| Antibody | Flag (rabbit monoclonal) | Cell Signaling Technology | 14793 | (1:1000) |
| Antibody | HA (rabbit monoclonal) | Cell Signaling Technology | 3724 | (1:1000) |
| Antibody | RHNO1 (rabbit polyclonal) | Novus | NBP1-93694 | Discontinued |
| Antibody | RHNO1 (rabbit polyclonal) | Sigma | HPA038682 | Discontinued |
| Antibody | TOPBP1 (rabbit monoclonal) | Cell Signaling Technology | 14342 | (1:1000) |
| Antibody | RAD9 (rabbit polyclonal) | Santa Cruz | sc-8324 | Discontinued |
| Antibody | RAD1 (rabbit monoclonal) | Invitrogen | 702149 | (1–2 µg/mL) |
| Antibody | HUS1 (rabbit monoclonal) | Cell Signaling Technology | 16416 | (1:1000) |
| Antibody | Anti-HA magnetic beads (mouse monoclonal) | Pierce | 88837 | |
| Cell line (*Homo sapiens*) | COV362 | Sigma | 7071904 | |
| Cell line (*Homo sapiens*) | COV318 | Sigma | 7071903 | |
| Cell line (*Homo sapiens*) | KURAMOCHI | JCRB | JCRB0098 | |
| Cell line (*Homo sapiens*) | OVSAHO | JCRB | JCRB1046 | |
| Cell line (*Homo sapiens*) | SNU-119 | Korean Cell Line Bank | 00119 | |

*Continued on next page*

*Appendix 1—key resources table continued*

| Reagent type (species) or resource | Designation | Source or reference | Identifiers | Additional information |
|---|---|---|---|---|
| Cell line (*Homo sapiens*) | OVCAR4 | NCI | | |
| Cell line (*Homo sapiens*) | OVCAR8 | NCI | | |
| Cell line (*Homo sapiens*) | OVCAR3 | ATCC | HTB-161 | |
| Cell line (*Homo sapiens*) | CAOV3 | Anirban Mitra | | |
| Cell line (*Homo sapiens*) | OVCAR5 | Anirban Mitra | | |
| Cell line (*Homo sapiens*) | UWB1.289 | Lee Zou | | |
| Cell line (*Homo sapiens*) | UWB1-SyR12 | Lee Zou | | |
| Cell line (*Homo sapiens*) | UWB1-SyR13 | Lee Zou | | |
| Cell line (*Homo sapiens*) | hOSE | ScienCell | 7310 | |
| Cell line (*Homo sapiens*) | FT190 | Ronny Drapkin | | |
| Cell line (*Homo sapiens*) | FT282-E1 | Ronny Drapkin | | |
| Cell line (*Homo sapiens*) | FT282 | Ronny Drapkin | | |
| Cell line (*Homo sapiens*) | IOSE-21 | Francis Balkwill | | |
| Cell line (*Homo sapiens*) | IOSE-121 | Nelly Auersperg | | |
| Cell line (*Homo sapiens*) | 283T | ATCC | CRL-3216 | |
| Cell line (*Homo sapiens*) | U2OS-DR-GFP (282C) | Jeremy Stark | | |
| Cell line (*Homo sapiens*) | OVCAR8-DR-GFP | Larry Karnitz and Scott Kaufmann | | |
| Chemical compound, drug | Berzosertib | SelleckChem | VE-822 | |
| Chemical compound, drug | Olaparib | SelleckChem | ABT-888 | |
| Chemical compound, drug | Hydroxyurea | Sigma | H8627 | |
| Chemical compound, drug | Carboplatin | Sigma | C2538 | |
| Chemical compound, drug | Etoposide | Sigma | E1383 | |
| Chemical compound, drug | Doxycycline | Sigma | D9891 | |
| Commercial assay or kit | Direct-zol RNA Purification Kit | Zymo Research | R2072 | |
| Commercial assay or kit | Comet Assay Kit | Trevigen | 4250-050-K | |

*Continued on next page*

*Appendix 1—key resources table continued*

| Reagent type (species) or resource | Designation | Source or reference | Identifiers | Additional information |
|---|---|---|---|---|
| Commercial assay or kit | iScript cDNA Synthesis Kit | Bio-Rad | 1708890 | |
| Commercial assay or kit | DNA-free kit | Ambion | AM1906 | |
| Commercial assay or kit | NE-PER Nuclear and Cytoplasmic Extraction Kit | Pierce | 78833 | |
| Commercial assay or kit | BCA protein assay | Pierce | 23225 | |
| Commercial assay or kit | Enhanced chemiluminescence | Pierce | 32106 | |
| Commercial assay or kit | Puregene Tissue Kit | QIAGEN | 158667 | |
| Commercial assay or kit | EZ DNA Methylation Kit | Zymo Research | D5001 | |
| Commercial assay or kit | QIAquick Gel Extraction Kit | QIAGEN | 28704 | |
| Commercial assay or kit | TOPO TA Cloning Kit | Invirogen | K457501 | |
| Commercial assay or kit | FirstChoice RLM-RACE Kit | Ambion | AM1700 | |
| Commercial assay or kit | Dual-Luciferase Reporter Assay System | Promega | E1910 | |
| Commercial assay or kit | Phospha-Light SEAP Reporter Gene Assay System | Thermo Scientific | T1015 | |
| Commercial assay or kit | TRIzol reagent | Thermo Scientific | 15596026 | |
| Commercial assay or kit | Lipofectamine 2000 reagent | Life Technologies | 11668019 | |
| Commercial assay or kit | QuickExtract | Lucigen | QE09050 | |
| Commercial assay or kit | M-PER | Pierce | 78501 | |
| Commercial assay or kit | Halt Protease and Phosphatase Cocktail | Pierce | 78440 | |
| Commercial assay or kit | Turbo Nuclease | Sigma | T4330-50KU | |
| Commercial assay or kit | AlamarBlue | Bio-Rad | BUF012A | |
| Commercial assay or kit | Vectashield with DAPI | Vector Laboratories | H-1200-10 | |
| Genetic reagent (*Homo sapiens*) | pCBASceI | Addgene | 26477 | Plasmid |
| Genetic reagent (*Homo sapiens*) | pCW57-GFP-P2A-MCS | Addgene | 71783 | Plasmid |
| Genetic reagent (*Homo sapiens*) | pCW57-GFP-P2A-MCS-(Neo) | Addgene | 89181 | Plasmid |
| Genetic reagent (*Homo sapiens*) | pCW57-MCS1-P2A-MCS2-(Hygro) | Addgene | 80922 | Plasmid |
| Genetic reagent (*Homo sapiens*) | pCW57-MCS1-P2A-MCS2-(Blast) | Addgene | 80921 | Plasmid |

*Appendix 1—key resources table continued*

| Reagent type (species) or resource | Designation | Source or reference | Identifiers | Additional information |
|---|---|---|---|---|
| Genetic reagent (*Homo sapiens*) | pCW57-RFP-P2A-MCS | Addgene | 78933 | Plasmid |
| Genetic reagent (*Homo sapiens*) | pCW57-RFP-P2A-MCS-(Neo) | Addgene | 89182 | Plasmid |
| Genetic reagent (*Homo sapiens*) | pTRIPZ Non-silencing | Dharmacon | NC0257175 | Plasmid |
| Genetic reagent (*Homo sapiens*) | pGL4 Luc Rluc Empty | Addgene | 64034 | Plasmid |
| Genetic reagent (*Homo sapiens*) | pCMV6-SEAP | Addgene | 24595 | Plasmid |
| Genetic reagent (*Homo sapiens*) | pTurbo-GFP-RFP-empty | Custom | | Plasmid |
| Genetic reagent (*Homo sapiens*) | pTurbo-GFP-PGK | Custom | | Plasmid |
| Genetic reagent (*Homo sapiens*) | pTurbo-RFP-PGK | Custom | | Plasmid |
| Genetic reagent (*Homo sapiens*) | pTurboGFP-RFP-FOXM1-RHNO1 | Custom | | Plasmid |
| Genetic reagent (*Homo sapiens*) | pLentiCRISPRv2 | Addgene | 52961 | Plasmid |
| Genetic reagent (*Homo sapiens*) | PX458-WT-Cas9-empty | Addgene | 48138 | Plasmid |
| Genetic reagent (*Homo sapiens*) | lenti-SAMv2-Puro | Addgene | 92062 | Plasmid |
| Genetic reagent (*Homo sapiens*) | lenti-MPHv2-Neo | Addgene | 92065 | Plasmid |
| Genetic reagent (*Homo sapiens*) | pLV hU6-sgRNA hUbC-dCas9-KRAB-T2A-Puro (KRAB) | Addgene | 71236 | Plasmid |
| Genetic reagent (*Homo sapiens*) | psPAX2 | Addgene | 12260 | Plasmid |
| Genetic reagent (*Homo sapiens*) | pMD2.G | Addgene | 12259 | Plasmid |
| Genetic reagent (*Homo sapiens*) | pOTB7-RHNO1 | Harvard PlasmID Repository | HsCD00326310 | Plasmid |
| Genetic reagent (*Homo sapiens*) | pENTR-MGC RHNO1 siR | Stephen Elledge | | Plasmid |
| Genetic reagent (*Homo sapiens*) | pENTR-MGC RHNO1 SWV | Stephen Elledge | | Plasmid |
| Genetic reagent (*Homo sapiens*) | pCMV6 AN-HA empty | Origene | PS100013 | Plasmid |
| Genetic reagent (*Homo sapiens*) | AdNGUS24i | Frank Graham and Phillip Ng | | Adenovirus |
| Software, algorithm | Prism 8 statistical software | GraphPad | | |
| Software, algorithm | Lasergene SeqMan Pro | DNASTAR | | |
| Software, algorithm | Comet Analysis Software | Trevigen | | |
| Software, algorithm | R package | R Project | | |
| Software, algorithm | Cell Ranger Single Cell Software | 10X Genomics | | |
| Software, algorithm | GSEA software version 3 | Broad Institute | | |
| Software, algorithm | Cufflinks (v2.1.1) | https://doi.org/10.1038/nbt.1621 | | |

*Appendix 1—key resources table continued*

| Reagent type (species) or resource | Designation | Source or reference | Identifiers | Additional information |
|---|---|---|---|---|
| Software, algorithm | Trim Galore software package | The Babraham Institute | | |
| Software, algorithm | TopHat (v2.0.8) | https://doi.org/10.1186/gb-2013-14-4-r36 | | |
| Software, algorithm | Cuffdiff (v2.1.1) | https://doi.org/10.1038/nbt.2450 | | |
| Software, algorithm | Perl | The Perl Foundation | | |
| Software, algorithm | scImpute | https://doi.org/10.1038/s41467-018-03405-7 | | |

