## [Decision Letter]

**Acceptance summary:**

The current manuscript by Barger et al. describes a bidirectional promoter (BDP) regulating FOXM1 and RHNO1. This paper was able to outline some of the putative functional ramifications of co-regulating FOXM1 and RHNO1 in ovarian cancer and illuminated a hitherto unappreciated role for RHNO1 in the regulation of DNA repair and resistance to therapy. These findings support future studies designed to interrogate and even target RHNO1 and its bidirectional promoter in cancers, with a particular emphasis on therapy resistant ovarian cancers.

**Decision letter after peer review:**

Thank you for submitting your article "Co-regulation and functional cooperativity of FOXM1 and RHNO1 bidirectional genes in ovarian cancer" for consideration by *eLife*. Your article has been reviewed by 3 peer reviewers, including Lynne-Marie Postovit as the Reviewing Editor and Reviewer #1, and the evaluation has been overseen by Maureen Murphy as the Senior Editor. The following individual involved in review of your submission has agreed to reveal their identity: Trevor G Shepherd (Reviewer #2).

The reviewers have discussed the reviews with one another and the Reviewing Editor has drafted this decision to help you prepare a revised submission.

Summary:

The current manuscript by Barger et al. describes a bidirectional promoter (BDP) regulating FOXM1 and RHNO1. Based on a very large number of well-described, appropriate and properly executed experiments, the authors concluded that FOXM1 and RHNO1 are coregulated in ovarian cancers and functionally cooperate in ovarian cancer cells. While this discovery describes a rather standard BDP, this paper was able to outline some of the putative functional ramifications of co-regulating FOXM1 and RHNO1. Specifically, this paper presented a previously unknown role for RHNO1 in ovarian cancers wherein the locus (containing the BDP) is located. Indeed, some of these functions may have previously been attributed to FOXM1 alone. This possibility was not fully explored in this manuscript, and instead a concept of co-operativity was introduced.

Three scientists, including myself, have reviewed your paper. We were all impressed with the work. The study is rigorous and the data are well organized and presented in a logical fashion. However, enhancements would be needed prior to publication in *eLife*. Specifically, studies are needed to better understand the cooperative role of RHNO and the BDP (versus FOXM1) in DNA repair and the susceptibility of ovarian cancer cells to PARP inhibitors.

Accordingly, please address all of the reviewer's concerns (outlined below). A certain number of experiments would also be required. These include an enhancement of the RNAseq analysis as well as experiments designed to better elucidate the role of RHNO in ovarian cancer DNA repair and/or to better characterize the concept of co-operativity. The referees discussed the need for in vivo models; and while it was felt that these would enhance the study, we agreed that these experiments would be out of the scope of the work and thus not needed for this paper.

Essential revisions:

1. The notion that FOXM1 and RHNO1 increase in cell lines in accordance with copy number (Figure 2E) is not overly convincing given the RT-PCR assay used and putative differences in PCR efficiencies. Please include absolute copy numbers for FOXM1 and RHNO1 (with ddPCR or a standard curve) and clearly demarcate copy numbers for each cell line analyzed. Also, Figure 2F- the authors state that protein expression increased with genomic copy number. This may be the case for SNU-119 but seems to be less obvious for the other cell lines. Please provide additional comments.

2. Several studies have demonstrated a role for FOXM1 in the regulation of ovarian cancer phenotypes including DNA repair in response to PARP inhibition, EMT and glucose metabolism. This is in contrast to the results obtained in this study, particularly since gene alterations detected with RNAseq seemed minimal. This could be due to the use of different cell lines and/or may have been somehow associated with the RNA sequencing analyses. This must be rectified. The following enhancements are suggested:

– The description of the RNAseq analysis should be improved. How many biological replicates were analyzed?

– Please include a heatmap and list of altered genes for each cell line.

– Please indicate the percent overlap in alterations between cell lines.

– Please include principle components analyses.

– Since it seems that RHNO1 is mediating a number of effects associated with the BDP, it would be important to determine the extent to which RHNO1, FOXM1 and/or BDP ablation elicits similar responses.

– The validations have no statistics, and there are no measurements of protein. It is difficult, with this data, to conclude that cell cycle genes are altered, as suggested.

3. The authors performed single cell (sc) RNA-seq in FTE and HGS cell lines but the results are confined to two observations in a table as unpublished data. These data overall are highly relevant to the current study and should be elaborated on in the paper, including figures of the clusters, descriptions and gene lists.

4. The cell cycle analyses in 5C, 7D and 8E do not show major changes. It is unclear if the cells were synchronized and statistics are absent. Given the aforementioned weaknesses in the RNAseq data, I would suggest removing this data. Otherwise, the experiments must be improved, with solid assays to measure proliferative indices, as well as an improved workflow for cell cycle analysis.

5. It seems that a putative role for RHNO1 in the regulation of DNA repair in ovarian cancer is a major discovery in this paper. Hence, the experiments showing this should be enhanced.

– RHNO1 localization studies should be expanded to include IF, perhaps in combination with γ-H2AX foci imaging.

– DNA repair should be directly measured in the cells used in Figures 7, 8 and 9. This would include all of the assays shown in S14 (Showing a role for RHNO1 in DNA repair) as well as a measurement of γ-H2AX foci. At least one model should also determine whether the effects of BDP KO can be rescued by RHNO1 (but not the SWV mutant). These experiments are needed to adequately link the BDP to the regulation of DNA repair (and susceptibility to PARP inhibitors) and to firmly establish the roles of FOXM1 and RHNO1 in DNA repair independently of proliferation which could be a confounder in the ALAMAR BLUE assay (which is an estimate of live cell numbers).

– Analysis of FOXM1 and RHNO1 in tumors from patients that were resistant to PARPi or developed resistance to PARPi would support the in vitro conclusions. Do the authors have access to such samples?

– In addition to PARPi, cellular data for response to platinum chemotherapy (sensitivity/resistance) would be informative particularly in the context of the disease. Have the authors performed similar experiments using platinum in the ovarian cancer cells?

6. In a number of experiments (Figure 5 and Figure 6) it is suggested that RHNO1 and FOXM1 work cooperatively. It is difficult to see how the data show this; as the effects of dual knockdown is only marginally greater than that of RHNO1 knockdown. This may not even be the case as it does not appear that multiple comparisons were made (for example with an ANOVA). Please conduct an ANOVA on Figure 5B, 5C (if it is kept), 6E, 6F, 7D (if it is kept), 8E (if it is kept), and S15D and adjust conclusions accordingly. Also, in Figure 6F – dual loss and HR efficiency in the bone cancer cell line was more robust than the ovarian cancer cell line, which showed only a modest although statistically significant difference. Did the authors examine additional HGS ovarian cancer line to support the cooperativity? Perhaps the authors can better define or focus this functional cooperativity, or consider describing it more accurately.

---

## [Author Response]

Essential revisions:1. The notion that FOXM1 and RHNO1 increase in cell lines in accordance with copy number (Figure 2E) is not overly convincing given the RT-PCR assay used and putative differences in PCR efficiencies. Please include absolute copy numbers for FOXM1 and RHNO1 (with ddPCR or a standard curve) and clearly demarcate copy numbers for each cell line analyzed. Also, Figure 2F- the authors state that protein expression increased with genomic copy number. This may be the case for SNU-119 but seems to be less obvious for the other cell lines. Please provide additional comments.

We show that *FOXM1* and *RHNO1* mRNA expression correlates with copy number in HGSC in RNA-seq and microarray data in Figure 3A (HGSC tumors, N=157; p<0.0001) and 3B (HGSC cell lines, N=23; p<0.0001). The data in Figure 3C-D only serve as a small validation using laboratory samples from cell lines. Figure 3C presents absolute RNA copy number data determined from standard curves of all target genes and is normalized to 18s rRNA, which reflects cDNA input. PCR efficiencies approach 100% for target genes. We revised Figure 3C-D to present the copy number for each cell line, as determined by the CCLE and NCI60 consortia. A direct relationship between genomic copy number and *FOXM1* and *RHNO1* expression is apparent, but additional mechanisms regulate their expression, as we have described for FOXM1 previously (Barger et al., Oncotarget, 2015; Barger et al., Cancers, 2019).

We added the following sentence to the text (p. 6): “Although the associations were significant, they were not uniform, indicating that mechanisms besides copy number status regulate FOXM1 and RHNO1 expression in HGSC. This is consistent with our prior studies of FOXM1 regulation (22,32).”

2. Several studies have demonstrated a role for FOXM1 in the regulation of ovarian cancer phenotypes including DNA repair in response to PARP inhibition, EMT and glucose metabolism. This is in contrast to the results obtained in this study, particularly since gene alterations detected with RNAseq seemed minimal. This could be due to the use of different cell lines and/or may have been somehow associated with the RNA sequencing analyses. This must be rectified. The following enhancements are suggested:– The description of the RNAseq analysis should be improved. How many biological replicates were analyzed?– Please include a heatmap and list of altered genes for each cell line.– Please indicate the percent overlap in alterations between cell lines.– Please include principle components analyses.– Since it seems that RHNO1 is mediating a number of effects associated with the BDP, it would be important to determine the extent to which RHNO1, FOXM1 and/or BDP ablation elicits similar responses.– The validations have no statistics, and there are no measurements of protein. It is difficult, with this data, to conclude that cell cycle genes are altered, as suggested.

We agree with this comment, and point out that presentation of bulk RNA-seq data was limited to one supplementary figure (Figure S13). After consideration of the additional studies required to characterize the impact of FOXM1 and/or RHNO1 knockdown and F/R-BDP repression on HGSC cell gene expression, we decided to remove bulk RNA-seq studies from the manuscript. Such studies are tangential to the point of the manuscript, are extensive, and are better suited for later presentation as a separate paper.

3. The authors performed single cell (sc) RNA-seq in FTE and HGS cell lines but the results are confined to two observations in a table as unpublished data. These data overall are highly relevant to the current study and should be elaborated on in the paper, including figures of the clusters, descriptions and gene lists.

We revised the manuscript to add new scRNA-seq data relevant to FOXM1 and RHNO1. Figures 4C-D show the normalized expression counts of *FOXM1* and *RHNO1* in FT282 and OVCAR8 cells, respectively, while Figure 4E shows the overall level of *FOXM1* and *RHNO1* expression in the two cell types. Figure 5 shows the distribution of correlation coefficients for all bidirectional gene (BDG) pairs in the two cell types and plots the correlations observed between *FOXM1* and *RHNO1*. Figure 5, Table Supplement 1, presents the raw correlation data for all genomic BDG pairs. Figure 5, Figure Supplement 1, shows principal component analysis (PCA) of FT282 and OVCAR8 transcriptomes. Full description of all scRNA-seq data, unrelated to BDGs and FOXM1/RHNO1, is beyond the scope of the manuscript. We deposited the scRNA-seq into GEO (GSE 150864).

4. The cell cycle analyses in 5C, 7D and 8E do not show major changes. It is unclear if the cells were synchronized and statistics are absent. Given the aforementioned weaknesses in the RNAseq data, I would suggest removing this data. Otherwise, the experiments must be improved, with solid assays to measure proliferative indices, as well as an improved workflow for cell cycle analysis.

We agree that the cell cycle changes observed in FOXM1 and RHNO1 depleted cells, in the absence of drug treatment, showed minor changes. Therefore, as suggested, we removed these data. After olaparib treatment, depletion of FOXM1 and/or RHNO1 did significantly alter OVCAR8 cell cycle (Figure 17D, Figure 18E). These analyses used asynchronous cell cultures. In the future, we plan to use cell synchronizations to better define cell cycle effects downstream of FOXM1 and/or RHNO1 depletion in HGSC cells.

5. It seems that a putative role for RHNO1 in the regulation of DNA repair in ovarian cancer is a major discovery in this paper. Hence, the experiments showing this should be enhanced.– RHNO1 localization studies should be expanded to include IF, perhaps in combination with γ-H2AX foci imaging.

IF of endogenous RHNO1 is not feasible due to the lack of a suitable primary antibody. We attempted IF of HA-RHNO1 but did not observe nuclear foci so we discontinued that line of studies. We conducted western blot analyses of γ-H2AX levels in chromatin from FOXM1 and/or RHNO1 knockdown OVCAR8 cells, with or without olaparib treatment (Figure 17C). γ-H2AX foci analyses gave inconsistent results so we are not comfortable reporting them at this time.

– DNA repair should be directly measured in the cells used in Figures 7, 8 and 9. This would include all of the assays shown in S14 (Showing a role for RHNO1 in DNA repair) as well as a measurement of γ-H2AX foci. At least one model should also determine whether the effects of BDP KO can be rescued by RHNO1 (but not the SWV mutant). These experiments are needed to adequately link the BDP to the regulation of DNA repair (and susceptibility to PARP inhibitors) and to firmly establish the roles of FOXM1 and RHNO1 in DNA repair independently of proliferation which could be a confounder in the ALAMAR BLUE assay (which is an estimate of live cell numbers).

The data shown in the prior Figure 7 and Figure S14 (now Figure 17 and Figure 13) are from OVCAR8 cells with dox-inducible FOXM1 and/or RHNO1 shRNAs. We presented functional HR data (DR-GFP assay) for these cells in Figure 16, which we feel addresses the main point. We also note that the data shown in the prior Figure 8 (now Figure 18) are from OVCAR8 cells expressing NT sgRNA or F/R-BDP-KRAB sgRNA. Thus, we attempted to make OVCAR8 F/R-BDP-KRAB, DR-GFP cells. Unfortunately, we were unsuccessful, due to the complexity of the cell engineering coupled with severe limitations on time and lab personnel. We would like to complete such experiments in the future, but feel that our manuscript is extensive enough at present to warrant publication.

– Analysis of FOXM1 and RHNO1 in tumors from patients that were resistant to PARPi or developed resistance to PARPi would support the in vitro conclusions. Do the authors have access to such samples?

We queried GEO and contacted a number of clinical collaborators. Unfortunately, we have not yet identified suitable data to address this important question. We will continue to query GEO and follow the literature in this area to address this question in the future.

– In addition to PARPi, cellular data for response to platinum chemotherapy (sensitivity/resistance) would be informative particularly in the context of the disease. Have the authors performed similar experiments using platinum in the ovarian cancer cells?

We added data showing the impact of FOXM1 and RHNO1 knockdown on OVCAR8 response to carboplatin (Figure 20). Depletion of either or both genes reduced clonogenic growth following carboplatin treatment.

6. In a number of experiments (Figure 5 and Figure 6) it is suggested that RHNO1 and FOXM1 work cooperatively. It is difficult to see how the data show this; as the effects of dual knockdown is only marginally greater than that of RHNO1 knockdown. This may not even be the case as it does not appear that multiple comparisons were made (for example with an ANOVA). Please conduct an ANOVA on Figure 5B, 5C (if it is kept), 6E, 6F, 7D (if it is kept), 8E (if it is kept), and S15D and adjust conclusions accordingly. Also, in Figure 6F- dual loss and HR efficiency in the bone cancer cell line was more robust than the ovarian cancer cell line, which showed only a modest although statistically significant difference. Did the authors examine additional HGS ovarian cancer line to support the cooperativity? Perhaps the authors can better define or focus this functional cooperativity, or consider describing it more accurately.

We agree that for several cell phenotypes the effect of RHNO1 knockdown is similar to the effects of dual knockdown. We now present results of ANOVA testing and adjusted the interpretation of our results accordingly. We added examination of the effect of dual FOXM1 and RHNO1 knockdown in the CAOV3 HGSC cell line (Figure 12, Figure Supplement 1), as suggested.

We revised the title of the manuscript to accurately describe the relationship between FOXM1 and RHNO1 function:

Original title: Co-regulation and functional cooperativity of FOXM1 and RHNO1 bidirectional genes in ovarian cancer.

Revised title: Co-regulation and function of FOXM1/RHNO1 bidirectional genes in cancer.